# Interactions between immune cell types facilitate the evolution of immune traits

Tania Dubovik[1,4,9], Martin Lukačišin[1,9], Elina Starosvetsky[1,4], Benjamin LeRoy[2,5], Rachelly Normand[1,6], Yasmin Admon[1,4], Ayelet Alpert[1,7], Yishai Ofran[1,3,8], Max G'Sell[2] & Shai S. Shen-Orr[1✉]

An essential prerequisite for evolution by natural selection is variation among individuals in traits that affect fitness[1]. The ability of a system to produce selectable variation, known as evolvability[2], thus markedly affects the rate of evolution. Although the immune system is among the fastest-evolving components in mammals[3], the sources of variation in immune traits remain largely unknown[4,5]. Here we show that an important determinant of the immune system's evolvability is its organization into interacting modules represented by different immune cell types. By profiling immune cell variation in bone marrow of 54 genetically diverse mouse strains from the Collaborative Cross[6], we found that variation in immune cell frequencies is polygenic and that many associated genes are involved in homeostatic balance through cell-intrinsic functions of proliferation, migration and cell death. However, we also found genes associated with the frequency of a particular cell type that are expressed in a different cell type, exerting their effect in what we term *cyto-trans*. The vertebrate evolutionary record shows that genes associated in *cyto-trans* have faced weaker negative selection, thus increasing the robustness and hence evolvability[2,7,8] of the immune system. This phenomenon is similarly observable in human blood. Our findings suggest that interactions between different components of the immune system provide a phenotypic space in which mutations can produce variation with little detriment, underscoring the role of modularity in the evolution of complex systems[9].

A key precondition for evolution by natural selection is the availability of suitable variation in natural populations. Early studies in evolutionary computation have shown that increased complexity also increases the probability that random mutations produce pleiotropic effects negatively affecting fitness[10]. In other words, complex systems have a greater potential for getting trapped in local fitness optima. Thus, for the Darwinian process of evolution through mutation and selection to work in complex biological systems, these systems need to have evolved evolvability—an architecture such that mutations are likely to result in more adaptive phenotypes[2].

The immune system is a complex system intimately engaged in maintaining homeostasis and the struggle against pathogens, which makes it a prime target for the process of natural selection. This is evidenced by the fact that genes with immune function are among the fastest-evolving genes in mammalian and avian genomes[3,11]. In animals, immunity is achieved through an interplay between different types of immune cells[12–15]. The immune system of different species, or even that of individuals of the same species, can thus differ not only in individual cell types[16,17] but also in the interaction between them[18–20]. The relative importance of these two contributions towards the evolvability of the immune system, however, is unclear.

The decisive factor for a system's evolvability is how genotypic variation maps onto phenotypic variation. To address the question of evolvability of immune traits, we focused on the genetic determinants of the relative frequencies of individual immune cell types, also known as immune profiles. We chose this trait because immune profiles are highly variable between individuals[5], are functionally important in both health and disease[21–25], and are to a large extent determined by genetic factors[4,26,27]. At the same time, genetic determinants of immune profile variation remain elusive. Whereas a recent study that included more than half a million participants successfully identified the association of more than a 1,000 genes with the frequencies of five, low-resolution white blood cell types (neutrophils, monocytes, lymphocytes, basophils and eosinophils)[28,29], studies looking at white blood cells with higher resolution, but conducted only on hundreds of individuals, identified only a small number of associated genes[22–24,30–33]. Approaches combining both higher cellular resolution and necessary sample size have only recently started to emerge[34], illustrating the need for alternative approaches.

Here we performed a genetic association study of immune cell profiles in the Collaborative Cross (CC), a panel of highly genetically diverse yet inbred mouse strains[6,35–37]. The CC exhibits wide phenotypic

[1]Department of Immunology, Faculty of Medicine, Technion – Israel Institute of Technology, Haifa, Israel. [2]Department of Statistics, Carnegie Mellon University, Pittsburgh, PA, USA. [3]Department of Haematology and Bone Marrow Transplantation, Rambam Health Care Campus, Haifa, Israel. [4]Present address: CytoReason, Tel-Aviv, Israel. [5]Present address: Nike, Beaverton, OR, USA. [6]Present address: Massachusetts General Hospital, Boston, MA, USA. [7]Present address: Department of Oncology, Rambam Health Care Campus, Haifa, Israel. [8]Present address: Haematology and Bone Marrow Transplantation Department and the Eisenberg R&D Authority, Shaare Zedek Medical Centre, Faculty of Medicine, Hebrew University, Jerusalem, Israel. [9]These authors contributed equally: Tania Dubovik, Martin Lukačišin. ✉e-mail: shenorr@technion.ac.il

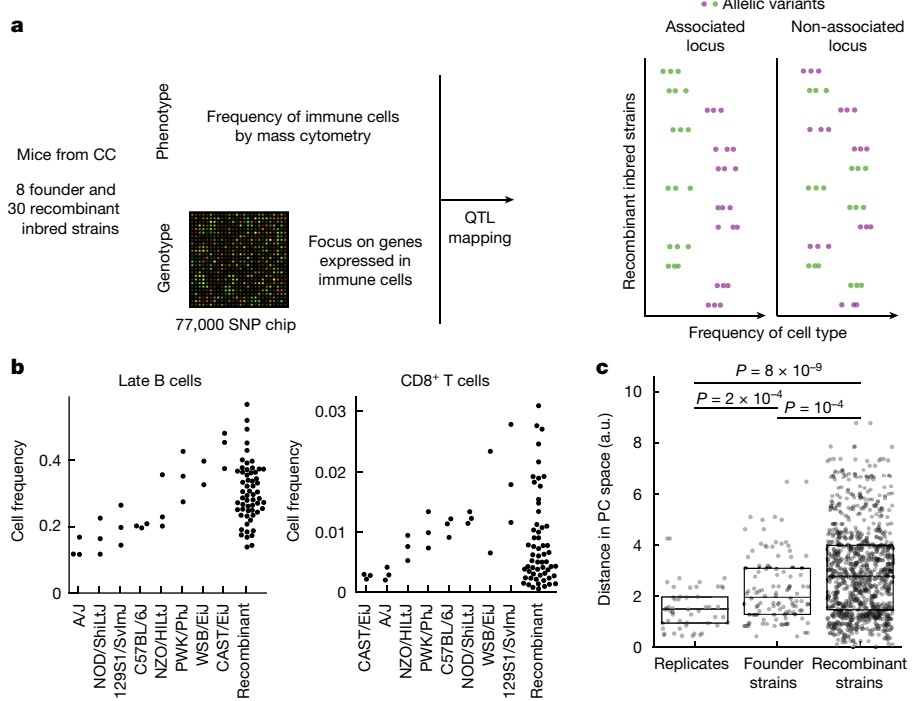

**Fig. 1 | Immune cell profiles are highly variable across CC mouse strains.**
**a**, Schematic of the experimental approach. The genome of CC recombinant strains was reconstructed at each locus based on single-nucleotide polymorphism (SNP) chip data and the genomes of the eight founder strains; immune cell frequencies were quantified by mass cytometry; the association of variants with immune cell type frequency was then quantified. **b**, Swarm-plot of frequencies of selected immune cell types from both CC founder strains and CC recombinant inbred strains used for the association study. Cell frequency is shown as a fraction of total live bone marrow cells. Founder strains are ordered according to median frequency for each cell subset; see Extended Data Fig. 1a for plots of all immune cell types assayed. **c**, Strip-plot of pairwise distances in principal component analysis (PCA) space between immune profiles within the respective categories. PCA was computed using the immune profiles for all measured animals that had all target cell types detected ($n = 42$); the Euclidean distance in the space defined by the first two PC (Extended Data Fig. 1b) is shown for all pairs within the respective categories. Boxplots denote median, first and third quartiles. $P$ values for two-sided $t$-test are shown. SNP chip created with Adobe Stock (https://stock.adobe.com).

variation in many functional immune traits[38–41], as well as in immune homeostatic composition[42,43], thereby allowing sufficient statistical power and reproducibility in a setting with minimal and equal environmental influences on immune cellular composition. We focused on immune cell profiles in the bone marrow, not readily accessible for association studies in humans. We associated and functionally analysed a large number of genes to immune cell abundance in a cell-specific manner. Finally we categorized exonic variants associated with immune cell frequency variation based on the respective genes being expressed or not in the respective immune cell type. Doing so enabled us to discover that those genes that are not expressed in the cell type whose variation they influence have accumulated variation under weak negative selection, supporting a role of intercellular interactions in immune system evolvability.

## CC mice vary widely in their immune profiles

To quantify variation in immune profiles in well-defined genetic backgrounds we leveraged mice from the CC panel[6]. The CC panel was previously designed to maximize heterogeneity between strains while maintaining individual homozygosity—eight founder mouse strains, including five laboratory and three wild inbred strains, had been crossed and then inbred to achieve homozygosity, for a total of about 100 strains[44]. Of those we profiled the bone marrow of eight founder strains in triplicate and 30 recombinant inbred strains in duplicate (Fig. 1a and Supplementary Table 1). To enable simultaneous profiling of multiple immune cell subtypes we subjected the extracted cells to profiling with time-of-flight mass cytometry (CyTOF). Using a broad panel of antibodies, we quantified the abundance of nine immune

cell populations—haematopoietic stem cells (HSCs), natural killer (NK) cells, CD8+ T cells, CD4+ T cells, total B cells, pro-B cells, late B cells, granulocytes and monocytes (for labelling and gating strategy see Methods, Supplementary Fig. 1 and Supplementary Tables 2 and 3). We observed a continuous range of immune cell frequencies among recombinant inbred strains, a typical phenotype of CC mice[42], with a dynamic range comparable to or even larger than that observed between the most extreme founder strains ($P = 10^{-4}$; Fig. 1b, Extended Data Fig. 1a and Supplementary Table 4). This is consistent with previously reported epistatic interactions[36]. Importantly for our study, variation between strains was much wider than that between mice of the same strain, allowing us to capture the genetic determinants of the variation with our study design (Fig. 1c and Extended Data Fig. 1b).

## Variation in immune profiles is polygenic

To link genotypic variation to phenotypic variation in immune profiles we associated the measured frequencies of different immune cell subpopulations with the known genetic profiles of the CC parental strains, which had all been previously sequenced[45]. To increase the interpretability of our association study we considered only genes with a potential function in the immune system as identified previously by the ImmGen consortium (a broad set of 7,965 genes)[46,47] and such that at least one founder strain contains an exonic variant in this gene, for a total of 6,902 genes represented by 15,458 loci (Methods). The exclusion of intronic and intergenic variants was driven by our goal to associate variants to specific genes[48]. For each of the nine immune cell subsets we applied a previously developed pipeline for quantitative trait loci (QTL) mapping in outbred mice[49], which determines the

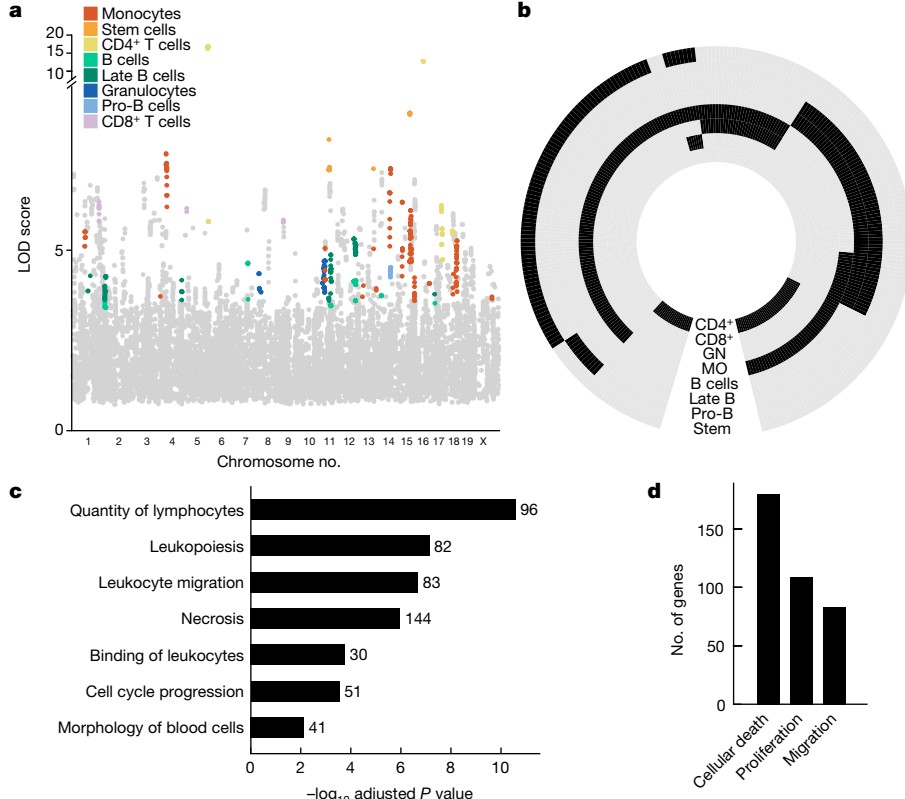

**Fig. 2 | Regulation of the frequency of immune cell types in bone marrow is polygenic and related to quantitative homeostatic balance. a**, Manhattan plot for associated loci. Coloured dots denote cell type with which genes were found associated based on significance threshold set according to FDR for each population separately; grey dots denote genes with an association below this threshold. Associated loci that did not pass validation by the second cohort are not shown. **b**, Circular heatmap of associations between genes and immune cell types. Black denotes association, grey denotes no association determined. Genes are clustered according to their association profile across cell types.

Cell types are indicated on each ring (CD4+, CD4+ T cells; CD8+, CD8+ T cells; GN, granulocytes; MO, monocytes); for NK cells no associations were determined. **c**, Functional enrichment of genes associated with the assayed immune traits as determined by ingenuity pathway analysis (IPA). Related functional terms were manually grouped and, for each group, the term with the lowest adjusted *P* value is shown; for details on grouping of terms see Extended Data Fig. 3b. One-sided Fisher's exact test with Benjamini–Hochberg correction was applied. **d**, Number of genes associated with selected homeostatic functions.

association of a given locus with the observed phenotype by taking into account the kinship between recombinant inbred strains—that is, the likelihood that a given strain contains the founder allele. In this way we identified genes associated with the frequency of one or more cell types with log odds ratio (LOD) greater than the 5% false discovery rate (FDR) threshold determined separately for each cell type by permutation analysis (Methods and Extended Data Fig. 2a). To ensure high stringency of our association study, we further measured immune cell profiles in a second, validation cohort of 48 mice stemming from 24 CC recombinant strains not included in the first cohort (Supplementary Tables 5 and 6). We performed analysis of variance to determine whether the variation in immune cell frequency in the validation cohort could be explained by allelic differences in each gene identified in the first cohort ($P_{adj}$ < 0.05), and retained genes only if the directionality of the effect was consistent across both cohorts. This stringent procedure resulted in 271 genes associated with high confidence with the frequency of one or more cell types in bone marrow (Fig. 2a). Due to the fact that we assayed functionally related traits, we leveraged signal propagation across traits to detect additional associations for this set of genes[50–52] (Methods). Overall we report 543 gene–cell type frequency associations across all assayed cell types (Fig. 2b and Supplementary Table 7).

To gain a biological understanding of the determinants of immune cell frequencies we performed functional enrichment analysis of trait-associated genes. The results suggested a role for genes involved in the cell-intrinsic functions of cellular movement, proliferation and

death (Extended Data Fig. 2c). To expand the resolution of this functional insight we increased the number of genes analysed for functional enrichment by relaxing the stringency of the association analysis for a total of 785 associated genes (Methods). Functional enrichment of this larger set confirmed the role of cell-intrinsic functions (Fig. 2c and Supplementary Table 8), with 31% of genes annotated for at least one of the functions of proliferation, cell death and cellular movement (Fig. 2d)—that is, basic determinants of homeostatic balance. Taken together, akin to the control of messenger RNA and protein abundance, cell subset abundance is also subject to turnover rates.

## Genes associated in *cyto-trans*

Because a considerable fraction of the genes that we discovered to be associated with immune cell frequencies are involved in cellular turnover, we wondered whether all variation in cell frequencies could be rationalized in terms of cell-intrinsic functions. In other words, we sought to identify genes that would influence the abundance of a cell type from outside of the given cell type, through direct or indirect interaction between the cell types. To explore this question we divided the associated genes into two groups (Fig. 3a)—those expressed in the cell type with whose abundance they are associated (henceforth *cyto-cis* genes) and those not expressed in the respective cell type (*cyto-trans* genes). To do this we took advantage of a comprehensive gene expression resource on sorted bone marrow cells from the Immunological Genome project[46], and the negative threshold for microarray signal

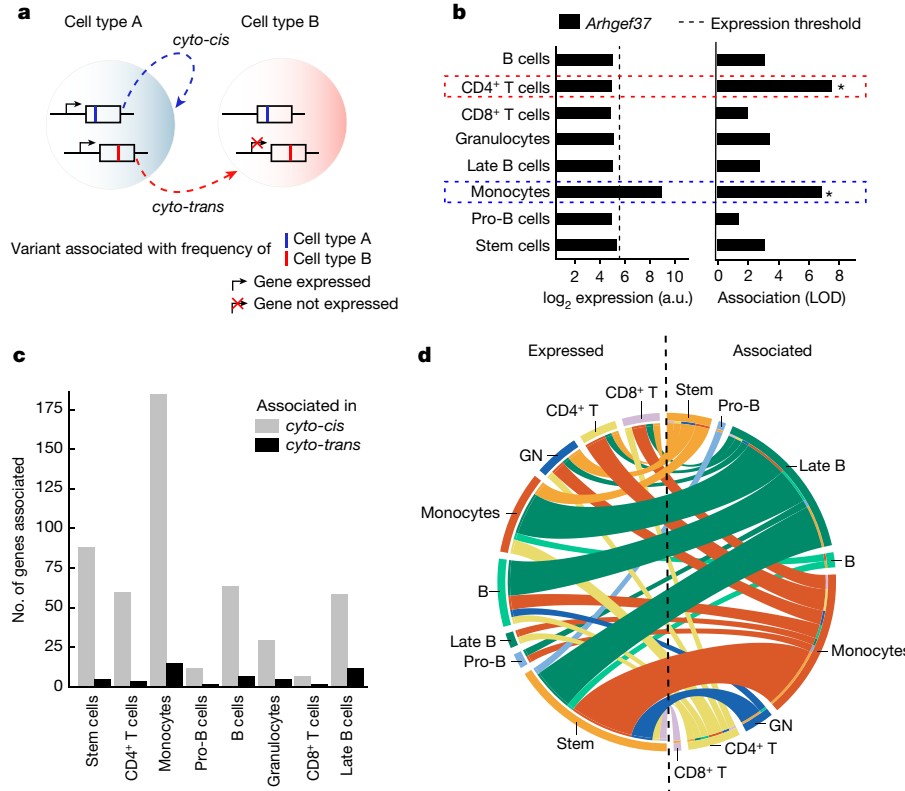

**Fig. 3 | Genes associated with frequency of specific immune cell types are frequently not expressed in the associated cell type. a**, Schematic of the definition of a genetic association acting in *cyto-cis* or *cyto-trans*. A gene associated in *cyto-cis* is expressed in the cell type of the phenotype with which it is associated; conversely, a *cyto-trans* gene is not expressed in the associated cell type but rather in a different cell type. **b**, Example of data used to classify associations as *cyto-cis* or *cyto-trans*. Gene expression was determined by ImmGen consortium, and LOD scores for association with immune cell frequencies are shown for the selected gene, *Arhgef37*. Black dashed vertical line indicates gene expression threshold used for binarization of gene

expression, and asterisks denote significant associations as determined by a permutation test (Methods). Coloured dashed rectangles highlight associations in *cyto-cis* and *cyto-trans*. **c**, Bar plot showing the number of genes associated in *cyto-cis* or *cyto-trans* for each respective cell type. Cell types are ordered by increasing proportion of *cyto-trans* associations. **d**, Split Circos plot depicting expression of genes associated with immune cell frequencies in *cyto-trans*. The colour of connecting lines is determined by the cell type with which the gene is associated; the colour of the rim on the left-hand side corresponds to the cell type in which the gene is expressed. For associated genes expressed in multiple cell types, multiple lines are shown. a.u., arbitrary units.

previously established by the consortium as corresponding to one per million reads in comparable RNA sequencing data[53] (dashed vertical line in Fig. 3b). As an example, *Arhgef37*, a guanine nucleotide exchange factor, was found to be associated with both monocyte and CD4+ T cell frequency (Fig. 3b, right). However, *Arhgef37* is expressed only in monocytes (Fig. 3b, left) and thus, with respect to monocyte frequency, *Arhgef37* acts in *cyto-cis* whereas, with respect to the frequency of CD4+ T cells, *Arhgef37* acts in *cyto-trans*.

Classifying all associations between genes and immune cell frequencies as either *cyto-cis* or *cyto-trans*, we observed a *cyto-cis* relationship for the majority of associations (91.9%, 499 associations comprising 257 genes; Fig. 3c). However, a non-negligible fraction of the associations (8.1%, 44 associations comprising 28 genes) observed across all cell types (Fig. 3c) demonstrated *cyto-trans* regulation—that is, these genes have the ability to regulate the abundance of at least one immune cell type without being expressed in that cell type. However, our analysis was restricted to exonic variants, which generally require expression to manifest their phenotype[54]. Consequently, the relevant immune cell types must be influenced by these variants through some form of interaction with cell types that express the protein, possibly mediated by ligands, metabolites, yet other cell types or in a connection between a cell and its precursor. Although many *cyto-trans* genes were expressed in HSCs, suggesting the control of abundance of a specific cell type from upstream in the differentiation lineage, overall *cyto-trans* genes

create a complex web of interactions between cell types (Fig. 3d). In addition, some *cyto-trans* associations were even shared across multiple cell types (Extended Data Fig. 3a). Compared with all associated genes, *cyto-trans* genes were enriched for a number of signalling pathways, congruent with their role in mediation of interactions between cell types (Extended Data Fig. 3b and Supplementary Table 9).

## Weaker negative selection in *cyto-trans*

Having identified the phenomenon of genes associated with the frequency of immune cell types in which they are not expressed, we next asked whether these *cyto-trans* genes have a special role in the evolution of immune traits. More specifically, we hypothesized that genes determining the frequency of the immune cell type but that are not expressed in that cell type, and thus cannot have any additional function in that cell type, might face weaker negative selection. To test this hypothesis we compared the evolutionary sequence conservation for *cyto-trans* versus *cyto-cis* genes as determined for the mouse immune system. To do so we used scores calculated with PhastCons[55], an established method used for quantification of evolutionary conservation based on a two-state hidden Markov model, across 60 vertebrate species including rodents, primates, carnivores and fish and spanning an evolutionary time of more than 500 million years[56] (Methods). The PhastCons model classifies each site as belonging to a conserved element or not,

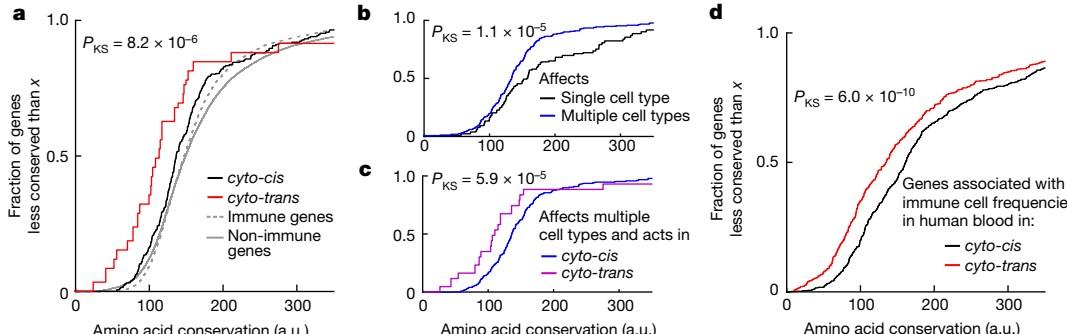

**Fig. 4 | Genes with *cyto-trans* associations facilitate the evolution of immune traits. a**, Empirical cumulative distribution functions for amino acid conservation scores of genes grouped by whether they were found to be associated with immune cell frequencies in only *cyto-cis* or also in *cyto-trans*. $P$ value for one-sided Kolmogorov–Smirnov test between these two groups is shown. Cumulative distributions of evolutionary conservation for immune genes considered in this study (Methods) and for non-immune genes in the mouse genome are shown for comparison in grey. **b**, As in **a**, except that genes are categorized according to whether they are associated with more than one immune cell type. **c**, As in **a**, except that only genes associated with frequencies of multiple immune cell types are considered. **d**, Empirical cumulative distribution functions for amino acid conservation scores of genes found to be associated with immune cell (neutrophil, monocyte, lymphocyte, basophil, eosinophil) frequencies in human blood[28,29]. Genes are grouped by whether they were found to be associated with immune cell frequencies only in *cyto-cis* or also in *cyto-trans* (Methods). $P$ value for one-sided Kolmogorov–Smirnov test is shown.

as well as assigning a score to each site representing the probability of negative selection of that site across evolution of the ensemble of species[55]. For each gene associated with at least one immune cell type abundance in our dataset, we calculated its mean conservation score by averaging the scores of all conserved sites located in its coding region. We observed that genes for which we found a *cyto-trans* association in our study were under significantly lower negative selection than those for which we found no *cyto-trans* association (one-sided Kolmogorov–Smirnov test, $P_{KS} = 8.2 \times 10^{-6}$; Fig. 4a).

Because *cyto-trans* regulation is, by definition, connected with interaction among different cell types, we wondered whether the recent evolution of a complex immune system in vertebrates had focused on coordinated changes in abundance across multiple immune cell types rather than just a single one. To test this hypothesis we classified genes to either (1) those associated with only a single cell type in our study (or with only closely related cell types within the same lineage; Methods) or (2) those associated with the frequency of multiple cell types, and quantified the evolutionary conservation as before. We indeed found that genes associated with the frequency of multiple cell types are evolutionarily less conserved than those affecting just one immune cell type ($P_{KS} = 1.1 \times 10^{-5}$; Fig. 4b).

We wondered whether the phenomena of genes driving evolution by eliciting coordinated changes across multiple cell types and evading negative selection by being involved in interactions between cell types are overlapping. *Cyto-trans* genes were not enriched among genes associated with multiple cell types ($P_{\chi2} = 0.61$) and, when considering only genes associated with multiple cell types, we observed that those with at least one *cyto-trans* association are under weaker negative selection ($P_{KS} = 5.9 \times 10^{-5}$; Fig. 4c). This suggests that genes involved in interactions between immune cell types have been an important source of variation, not merely because they would be more likely to affect multiple cell types but probably because they face fewer constraints related to their function.

The principle that *cyto-trans* genes promote evolvability cannot be specific to mice but, rather, should be true of any sufficiently advanced cell-based immune system. We therefore sought to validate our result using a larger experimental dataset from a different organism. To do so we analysed previously published genetic associations for the frequencies of immune cell types in human blood (1,046 genes in total)[28,29] and juxtaposed them with gene expression data from the Human Protein Atlas[57]. Here too we found that genes associated with cell types in which they were not expressed (*cyto-trans*, 358 genes) were

significantly less conserved than those expressed in the associated cell type ($P_{KS} = 6.0 \times 10^{-10}$; Fig. 4d, Extended Data Fig. 4 and Supplementary Table 11).

## Discussion

Natural selection as an evolutionary mechanism can act only on pre-existing population heterogeneity, which also needs to be heritable. The genetic contribution towards interindividual variation is thus a key variable determining evolvability. In the present study we leveraged heterogeneity in recombinant mice stemming from well-defined lineages to infer those genes influencing immune cell frequencies, an immune trait important in both health and disease[5,21–25], and juxtaposed these genes with the vertebrate evolutionary record to infer the determinants of immune system evolvability.

We found 271 genes with variants that contribute towards variation in the frequencies of various immune cell types. Although many of these do so through the cell-intrinsic functions of cell differentiation, proliferation and death, we also found that about 10% of the associated genes affect the frequencies of those cell types in which they are not expressed. These genes must be doing so in what we term *cyto-trans*— that is, via expression in another cell type, through direct or indirect interaction with the cell type in question. Notably, we found that the coding sequences of *cyto-trans* genes have been under weaker negative selection during vertebrate evolution than those of the genes found acting in *cyto-cis* only. This implies that the genetic determinants of interactions between different immune cell types are, when mutated, more amenable to producing near-neutral variation in immune cell frequencies compared with genes involved in internal cell regulation. In addition, as a separate phenomenon, we found that the same is true for genes associated with, and potentially coordinating, frequencies of multiple cell types. This earmarks genes involved in interactions between immune cell types as an important source of near-neutral variation.

Variation with strong positive or negative fitness effects has an appreciated role in the evolution of biological species, but the importance of near-neutral variation has only lately emerged as an important consideration for the capacity of biological systems to evolve. Recent work on transcription factor and protein structure evolution[2,7,58] has found that increased capacity for near-neutral variation—that is, increased mutational robustness—paradoxically facilitates evolution by supporting genetic diversity. This has solved the apparent conflict between

evolvability and robustness[8]—the seemingly contradictory requirement that phenotypes be significantly altered by genetic changes to allow selection of fitter phenotypes but not be altered so much as to undergo frequent mutations without harm. Our results offer an appealing extension of this evolvability principle in the context of the immune system—an enhancement of near-neutral genetic diversity is achieved through modularity. Conceptually one can speculate that, whereas the genetic determinants of inner cell life might have been arranged early in evolution to ensure functioning of individual immune cell types, the interaction between different immune cell types provides a phenotypic space in which mutations can continue to produce variation with little detriment.

The cell, the atomic unit of life, is well suited to correspond to a module in the evolutionary process of the immune system, consistent with *cyto-trans* genes acting as an important source of evolutionary novelty. Nevertheless, our observation that genes associated with frequencies of multiple cell types, and thus possibly coordination among them, are also an important source of evolvability irrespective of the *cyto-trans* phenomenon suggests the existence of multiple layers of modularity. Such staged modular design, in which modules are iteratively combined into yet more complex modules, has previously been proposed as one of the unifying features of evolved complex systems, including those of human origin[9]. Although evolvability through modularity is not expected to be limited to the immune system, due to the high need for its rapid evolution, the immune system is poised to manifest modularity exceptionally strongly. Further research into immune system evolvability thus could not only enlighten the design principles behind immune responses but also contribute to biomimetic solutions—for example, in the system-of-systems approach to engineering, which is similarly based on interactions between functional units[59]. Our work thus acutely demonstrates the need to bring the study of evolvability through modularity, previously explored with success in the context of protein structure[60], gene regulation[61] and metabolism[62], to complex cellular systems such as the immune system.

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

# Methods

## Mouse strains and sample collection

The CC founder strain males ($n$ = 3 per strain) of laboratory strains A/J, C57BL/6 J, 129S1Sv/ImJ, NOD/ShiLtJ and NZO/H1LtJ, and of wild-derived strains CAST/EiJ, PWK/PhJ and WSB/EiJ, were purchased from The Jackson Laboratory, delivered to the Systems Genetics Core Facility at The University of North Carolina (UNC) and killed at 6–8 weeks. CC recombinant mice were purchased from, handled, bred and killed by the Systems Genetics Core Facility at UNC. All procedures involving animals were performed according to the Guide for the Care and Use of Laboratory Animals, with previous approval by the Institutional Animal Care and Use Committee within the Association for Assessment and Accreditation of Laboratory Animal Care-accredited programme at UNC at Chapel Hill (Animal Welfare Assurance no. A-3410-01). In total we profiled 129 mice aged 8–14 weeks from 54 different complete lines, at least two mice per strain (Supplementary Table 1). Bone marrow and blood tissues were collected from necropsy following killing by $CO_2$. Bone marrow was flushed with a cold-cell staining medium (DPBS + 0.5% bovine serum albumin) from the femur and tibia using a 27.5-gauge needle and 10 ml syringe to achieve a single-cell suspension. Red blood cells from blood samples were eliminated by lysis buffer (BD Biosciences, catalogue no. 555899).

## Cell staining

Primary conjugates of mass cytometry antibodies were prepared using the Maxpar antibody conjugation kit (Fluidigm Inc.) according to the manufacturer's protocol (no. PRD002, Fluidigm), and optimal concentration was determined by titration according to the manufacturer's protocol. Cells from each sample were washed twice with Cell Staining Medium (Maxpar) and a total of 3 million cells were used for extracellular staining. Cells were resuspended in 500 µl containing 1:2,000 rhodium DNA intercalator (Fluidigm) for 20 min of live/dead cell staining. Samples were washed with Cell Staining Medium and resuspended in a total of 100 µl of metal-tagged antibody mix for 1 h for cell surface marker staining. Cells were then fixed with 1.6% paraformaldehyde (Sigma-Aldrich) in a total volume of 200 µl and stored at 4 °C. Cells were centrifuged, paraformaldehyde removed and iridium DNA intercalator staining was performed for 20 min at 1:2,000 dilution in a 500 µl volume to differentiate cells from debris. Finally, fixed samples were washed three times with deionized water immediately before data acquisition.

## Data acquisition and analysis

Samples were acquired using a CyTOF1 machine (DVS Sciences) at 500 events s$^{-1}$ for a total of 100,000–200,000 events per sample. Internal metal isotope bead standards were added for sample normalization as described previously[63]. Acquired data were uploaded to a Cytobank web server (Cytobank) for data processing and gating out of dead cells and normalization beads. To account for intrarun declines in mean marker intensity over time, the acquisition records were truncated at the moment when any of the channels drifted by more than 5% from its mean since the start of acquisition. We noted that, for several strains (maximum of eight per cell type), some cell type markers were not observed, probably due to allelic variants being low affinity to the respective antibody, consistent with previous literature[64], barring the direct use of unsupervised clustering approaches for cell subset characterization and quantification. We thus manually gated all samples (see Supplementary Fig. 1 for gating scheme) to estimate the frequency of nine major cell types across mouse strains. Mouse strains for which an antibody required to identify a cell subset was presumed low affinity were not included in the analysis of genetic associations for that cell subset (see Supplementary Tables 3 and 5 for exclusions). Resulting cell counts were then exported, adjusted to the total number of cells in the sample and are summarized in Supplementary Tables 4 and 6.

## Initial mapping of associated genetic loci

**Filtering of loci.** The CC mice were genotyped using MegaMUGA SNPchip with about 78,000 markers[36,65,66]. To reduce the extent of multiple hypothesis testing we chose to focus on genomic loci containing immune-related genes with genetic variants. To do so we included only loci that passed both of the following criteria: (1) those located within a broad set of genes identified previously by the ImmGen consortium (7,965 genes) as having a potential function in the immune system[46,47] and (2) those for which at least one of the CC founder strains harboured an exonic sequence variant. Using these two filtering criteria, we reduced the number of loci for testing from 77,725 to 15,458 located within 6,902 genes.

**Genotype–phenotype association.** We performed QTL mapping using the DOQTL R package[49]. In the first step, the haplotype of each recombinant mouse was reconstructed by per-locus estimation of the likelihood of it stemming from each of the founder strains. In the next step we searched for significant association between immune phenotype and each genomic locus, using an additive haplotype model that takes into account the kinship between recombinant CC mouse strains. To check for stability of observed associations we used a leave-one-out approach, always leaving one sample out and recalculating the strength of association; we retained only loci showing an association across all leave-one-out options according to the DOQTL procedure. Then, for each cell type we identified the significant association threshold as one that corresponds to 5% FDR (Extended Data Fig. 3a). We determined FDR by permutation analysis—repeatedly shuffling the labels of recombinant mouse strains (replicates were always assigned identical labels)—and calculated the strength of the cell type association.

## Validation of gene–trait association

We used a second cohort of mice to validate our findings. We determined the frequencies of nine immune cell subsets as in the exploratory cohort. Each significant gene–phenotype association identified in the first cohort was tested for significance in the second. We used analysis of variance to determine whether the variation in immune cell frequencies could be explained by allelic differences in mice in the validation cohort. Only associations with $P_{adj}$ < 0.05, following correction for multiple hypothesis testing using the Benjamini–Hochberg approach, were retained. Next we required that the directionality of the phenotype–genotype association be consistent between the first and second cohorts. We calculated Spearman correlation between the first and the second cohorts for the mean of the given phenotype stratified by allele; only genes with correlation greater than 0 were retained for further analysis.

## Signal propagation

Considering all genes that were validated to be associated with at least one assayed trait, we clustered them according to their association scores across all cell types using $k$-means clustering. The optimal number of clusters was determined using the silhouette method. For each cell type we determined associated clusters using their median association LOD scores. We considered a cluster as being associated with a given cell type frequency if the median LOD score of genes in the cluster for the given cell type was at least 40% of the highest median calculated for that cluster. This 40% threshold was chosen empirically based on the location of the minimum in the distribution of the relative LOD scores for gene cluster–cell type pairs, as shown in Extended Data Fig. 2b. The clusters were used solely for the purpose of signal propagation and not for any other analysis in this work.

## Gene enrichment analysis

We performed gene enrichment analysis on the combined list of associated genes for all cell subtypes with IPA (Qiagen) software (build

'ing_neptunite' 2023-11-19), with the Core Analysis module for 'Diseases & Functions', using all genes considered in the association study as the background set. Biological filters were set such that only mouse genes were considered, and only immune-related terms: disease-related terms were manually filtered out and the remaining functional terms manually grouped into higher-level functions. We similarly performed gene enrichment analysis on a larger list of genes after relaxing the stringency of the association test so as to have no requirement for the directionality of effect to be conserved in the validation cohort ($P_{adj} < 0.1$). Gene enrichment analysis of *cyto-trans* genes was performed similarly using IPA, with the Core Analysis module for 'Pathways' using all associated genes as the background set.

### Cyto-cis and cyto-trans determination

We defined the expression of a particular gene in a particular cell type based on ImmGen Consortium expression data[46] (GSE15907). We mapped between the populations defined by ImmGen and those profiled in our study (see Supplementary Table 10 for mapping). In the case where more than one ImmGen population was included in our cell subset definition, we required that at least one subpopulation express the gene of interest. To define each gene as expressed or not, we used an expression threshold defined by the ImmGen Consortium[53]. In brief, the consortium determined a threshold to consider a gene expressed by an empirical Gaussian mixture model, used to decompose the expression histograms into two Gaussian distributions. Low distribution was assumed to correspond to background noise, and the higher to the true signal. These yielded a conservative threshold for expression from the higher-expression Gaussian (above 120 a.u., 95% or greater probability of expression) and a threshold for the gene being silent from the lower-expression Gaussian (below 47 a.u., 95% or greater probability of the gene being silent). The range 47–120 a.u. was an intermediate range within which the gene might or might not be expressed. To be conservative towards calling *cyto-trans* based on the non-expression of the gene in the associated cell type, we considered the low ImmGen microarray threshold for gene expression of 47 a.u.

### Evolutionary conservation

We obtained PhastCons[55] conservation scores from the UCSC Genome Browser for 60 vertebrate species. Then, for each gene we calculated its conservation score by averaging the scores of all conserved elements located in its coding region. For the purposes of classification of genes as being associated with single or multiple cell types, all subtypes of B cells were considered a single cell type, as were granulocytes and monocytes.

### Validation using human data

To validate in a different species that genes associated with immune traits in *cyto-trans* exhibit lower evolutionary conservation, we obtained published data from two human studies of immune cell frequencies in blood[28,29]. Both studies measured the abundance of five immune cell types: basophils, eosinophils, lymphocytes, monocytes and neutrophils. In total, 1,046 genes were found to be associated with abundance of at least one of these cell types.

Gene expression data for human immune cells were obtained from the Human Protein Atlas[57] (http://www.proteinatlas.org). For lymphocytes, all genes expressed in both B and T cells were included. All genes associated with a cell type in which they were not expressed were considered as being associated in *cyto-trans*. We estimated conservation of the associated human genes in a similar manner as before for mouse genes ('Evolutionary conservation') using PhastCons[55] conservation scores for human genes considering 100 vertebrate species, downloaded from the UCSC Genome Browser.

### Reporting summary

Further information on research design is available in the Nature Portfolio Reporting Summary linked to this article.

### Data availability

The raw data files generated in CyTOF runs are available on community.cytobank.org (experiment nos. 116506 and 116507; first and second cohort, respectively). Mouse cell-specific gene expression was estimated using data generated by the ImmGen consortium, ImmGen Microarray Phase 1, GSE15907. Human cell-specific gene expression data were obtained from the Human Protein Atlas (https://www.proteinatlas.org/about/download).

### Code availability

The analysis code is available on GitHub (https://github.com/shenorrLabTRDF/CCanalysis).

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

**Acknowledgements** We thank F. P. M. de Villena, D. Miller and G. Shaw for providing us with the CC mouse strains and their help with the mouse experiments; A. Rolls, T. Ben-Shannan, H. Azulay-Debby, B. Korin and M. Schiller for help with mouse work and manuscript revision; A. Ziv-Kennet for help with data preprocessing; and F. Iraqi for initial set-up of the system. We thank J. Tsang, M. Choder, A. Segre, D. Melamed, G. Atzmon, M. Lukačišinová, B. Perets and members of the Shen-Orr laboratory for fruitful discussions. This study was supported through generous support from the Israel Science Foundation (grant no. 1365/12 and 1626/20), Anonymous Foundation, MALAT and the Colleck Research Fund. M.L. was supported by the Rubenstein-Technion Integrated Cancer Center Fellowship and the Aly Kaufman Fellowship. We thank Y. Abraham for his contribution to the design and creation of the schematic in the original version of Fig 1a.

**Author contributions** T.D., E.S. and S.S.S. carried out study design. T.D., M.L. and S.S.S. were responsible for conceptualization. Investigation was the responsibility of T.D., M.L., B.L., R.N., Y.A., A.A., Y.O. and M.G. Experimentation was carried out by T.D. and E.S. Data analysis was performed by T.D. and M.L. T.D., M.L. and S.S.S. wrote the manuscript draft. Writing, review and editing were carried out by T.D., M.L. and S.S.S. S.S.S. supervised and acquired funding.

**Competing interests** S.S.O. holds equity and is a consultant of CytoReason and holds an unpaid position with the Human Immunome Project. T.D., E.S. and Y.A., are employees and hold equity in CytoReason. R.N holds equity in CytoReason.

**Additional information**
**Correspondence and requests for materials** should be addressed to Shai S. Shen-Orr.

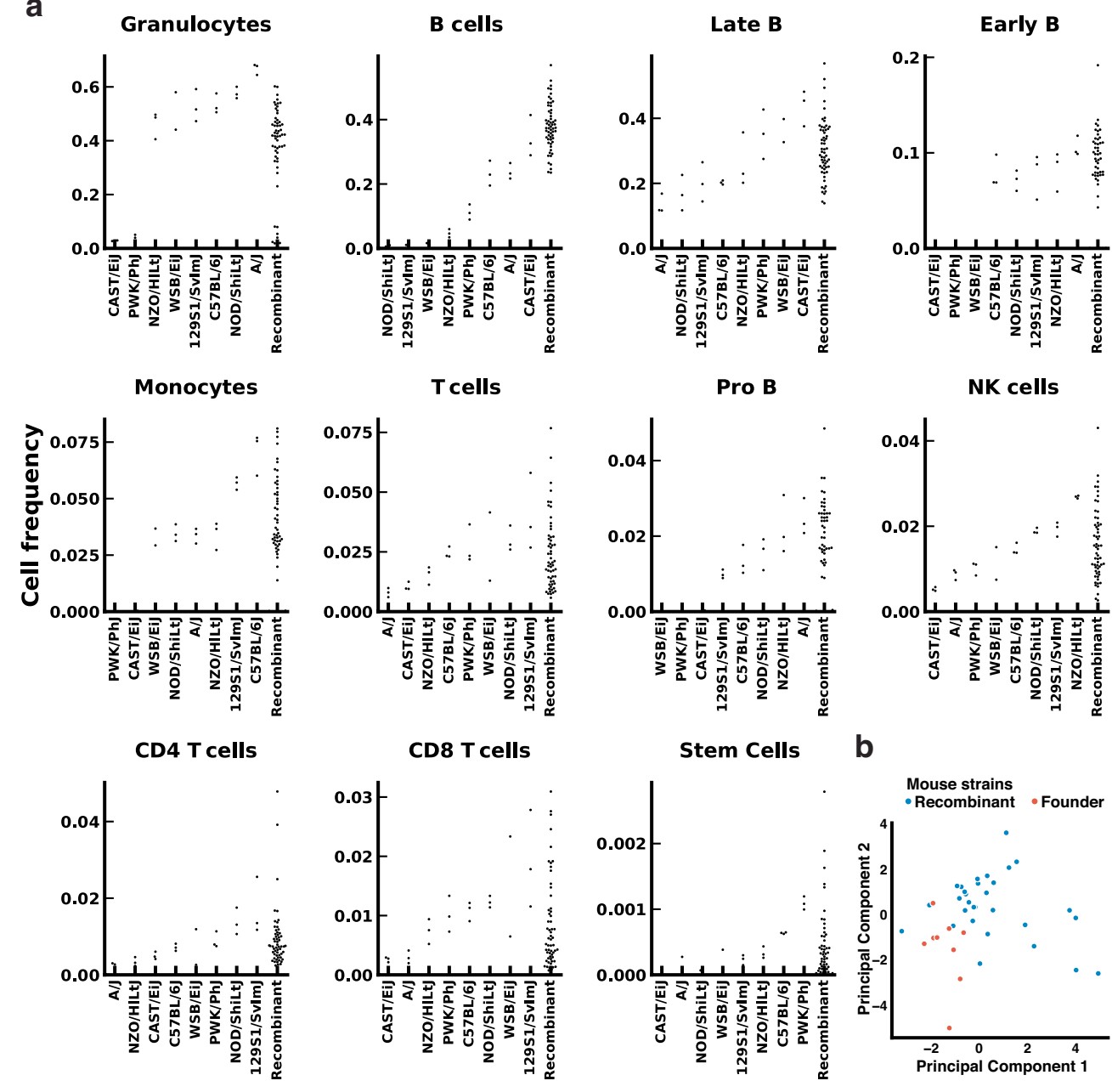

**Extended Data Fig. 1 | Immune cell frequencies of CC parent and recombinant strains used in the exploratory part of the association study. a.** Swarm plots of bone marrow immune cell frequencies, profiled using mass cytometry. Parent strains are ordered according to the median value for each of the immune cell subsets. **b.** Principal component analysis using immune cell frequencies of all mice measured in the first cohort, for which all target cell types could be detected. The first two principal components are shown.

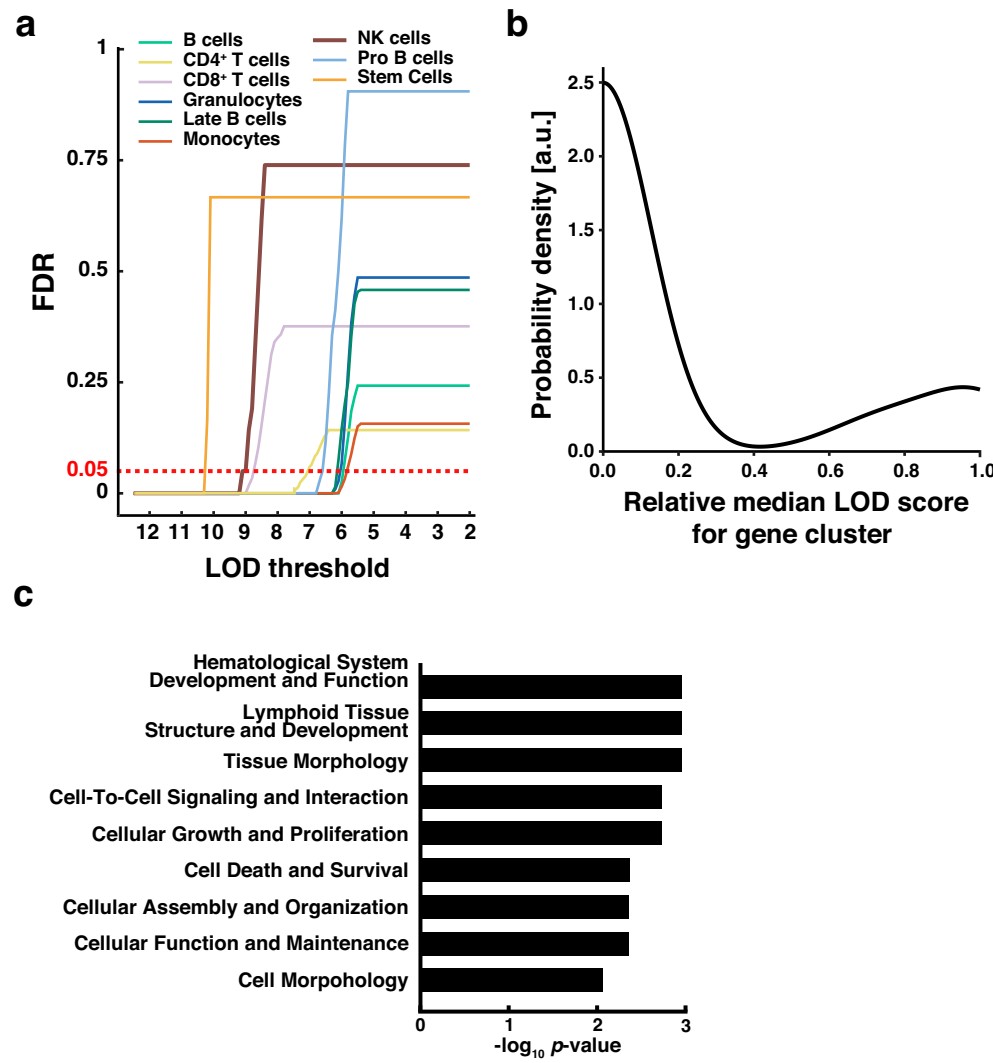

**Extended Data Fig. 2 | Genetic associations of immune cell frequencies.**
**a**. False discovery rate (FDR) as a function of threshold for association as inferred from permutation analysis (*Methods*) for each cell type. Red dashed line indicates the FDR of 5% used to determine the cell type-specific LOD score threshold used to find associated genes in the exploratory cohort. **b**. Distribution of relative median LOD scores for gene clusters during signal propagation. Genes found significantly associated to at least immune cell type were clustered based on their pattern of association across all the assayed cell types. For each cluster, median LOD score for each cell type was calculated and then divided by the maximum value of this quantity across all cell types. The distribution of the resulting values is plotted, with two clear peaks – all gene cluster-cell type pairs with median LOD score above 0.4 were considered associated. **c**. Bar-plot of functionally enriched categories. Functional groups that were found enriched using Ingenuity Pathway Analysis; terms with p < 0.05 are shown. x-axis indicates -$\log_{10}$ of *p*-value from one-sided Fisher's exact test.

**a**

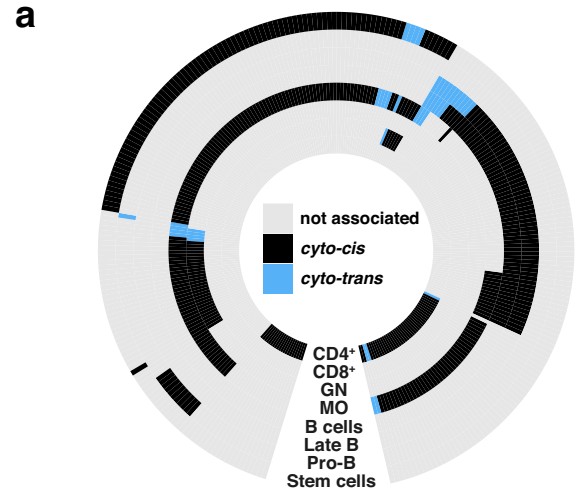

not associated
*cyto-cis*
*cyto-trans*

CD4⁺
CD8⁺
GN
MO
B cells
Late B
Pro-B
Stem cells

**b**

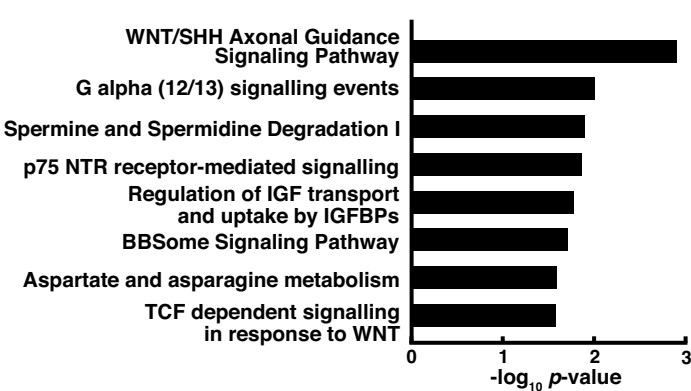

**c**

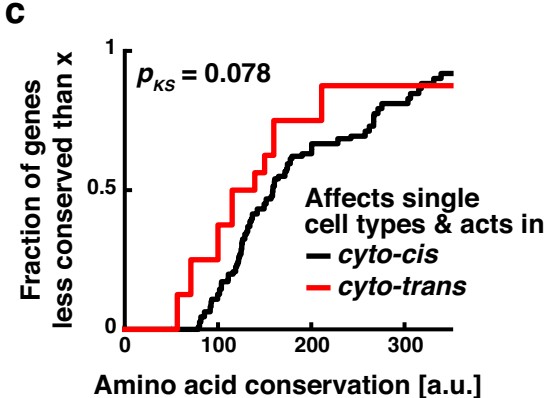

**Extended Data Fig. 3 | Analysis of *cyto-trans* associations. a**. Circular heatmap of associations between genes and immune cell types, classified according to expression in the respective cell type. Genes are clustered according to their association profile across cell types. Cell types are indicated on each ring (CD4⁺ – CD4⁺ T cells, CD8⁺ – CD8⁺ T cells, GN – Granulocytes, MO – monocytes). **b**. Bar-plot of functionally enriched categories in *cyto-trans* genes relative to all associated genes. Functional groups that were found enriched using Ingenuity Pathway Analysis (see Methods for details), terms with p < 0.05 are shown. x-axis indicates -log₁₀ of *p*-value from one-sided Fisher's exact test. **c**. Empirical cumulative distribution functions for amino acid conservation scores of genes found associated with a single immune cell type in this study, grouped by whether they were found associated in *cyto-cis* or in *cyto-trans*. *P*-value for one-sided Kolmogorov-Smirnov test between these two groups is stated.

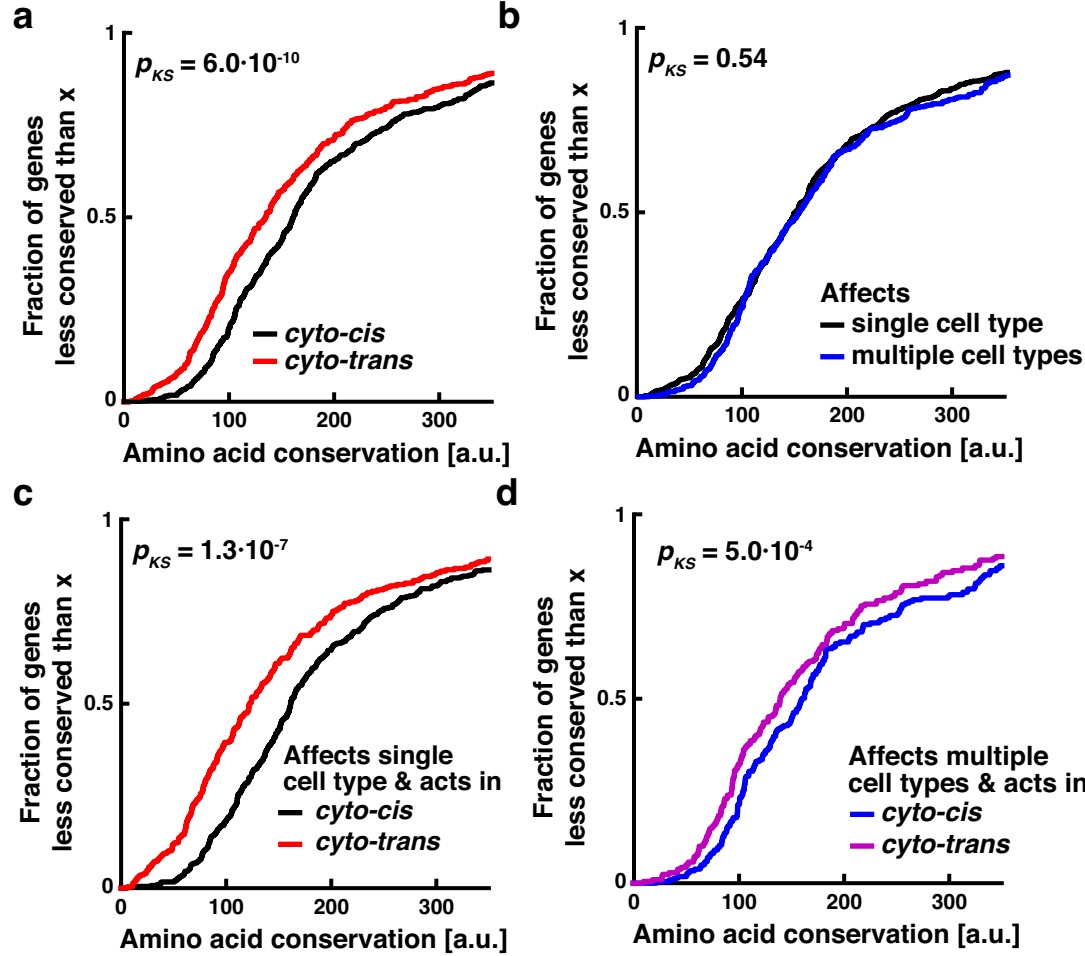

**Extended Data Fig. 4 | Evolutionary conservation of genes associated with immune cell frequencies in the human blood. a**. As in Fig. 4d. Empirical cumulative distribution functions for amino acid conservation scores of genes found associated with immune cell frequencies in the human blood by previous studies. Genes are grouped by whether they were found associated with immune cell frequencies only in *cyto-cis* or also in *cyto-trans* (*Methods*). **b**. As in (**a**), except genes are categorised according to whether or not they are associated to more than one immune cell type or not. **c**. As in (**a**), except only genes associated with frequencies of a single immune cell type are considered **d**. As in (**a**), except only genes associated with frequencies of multiple immune cell types are considered. *P*-values for one-sided Kolmogorov-Smirnov test are stated.

# Reporting Summary

## Statistics

For all statistical analyses, confirm that the following items are present in the figure legend, table legend, main text, or Methods section.

| n/a | Confirmed | |
|---|---|---|
| ☐ | ☒ | The exact sample size (*n*) for each experimental group/condition, given as a discrete number and unit of measurement |
| ☐ | ☒ | A statement on whether measurements were taken from distinct samples or whether the same sample was measured repeatedly |
| ☐ | ☒ | The statistical test(s) used AND whether they are one- or two-sided<br>*Only common tests should be described solely by name; describe more complex techniques in the Methods section.* |
| ☐ | ☒ | A description of all covariates tested |
| ☐ | ☒ | A description of any assumptions or corrections, such as tests of normality and adjustment for multiple comparisons |
| ☐ | ☒ | A full description of the statistical parameters including central tendency (e.g. means) or other basic estimates (e.g. regression coefficient) AND variation (e.g. standard deviation) or associated estimates of uncertainty (e.g. confidence intervals) |
| ☐ | ☒ | For null hypothesis testing, the test statistic (e.g. *F*, *t*, *r*) with confidence intervals, effect sizes, degrees of freedom and *P* value noted<br>*Give P values as exact values whenever suitable.* |
| ☒ | ☐ | For Bayesian analysis, information on the choice of priors and Markov chain Monte Carlo settings |
| ☒ | ☐ | For hierarchical and complex designs, identification of the appropriate level for tests and full reporting of outcomes |
| ☒ | ☐ | Estimates of effect sizes (e.g. Cohen's *d*, Pearson's *r*), indicating how they were calculated |

*Our web collection on statistics for biologists contains articles on many of the points above.*

## Software and code

Policy information about availability of computer code

| Data collection | CyTOF 1 acquisition software (v1) was used to collect the raw data from CyTOF machine. |
|---|---|
| Data analysis | Cytobank ( >= 3.2.1) was used to upload , explore and manually gate the fcs files.<br>Enrichment analysis was done using the IPA Spring release 2016 (Qiagen).<br>RStudio ,<br>Jupiter Notebook,<br>Custom script was used for all the analytical steps is available on Github : https://github.com/shenorrLabTRDF/CCanalysis |

For manuscripts utilizing custom algorithms or software that are central to the research but not yet described in published literature, software must be made available to editors and reviewers. We strongly encourage code deposition in a community repository (e.g. GitHub). See the Nature Portfolio guidelines for submitting code & software for further information.

## Data

Policy information about availability of data

All manuscripts must include a data availability statement. This statement should provide the following information, where applicable:

- Accession codes, unique identifiers, or web links for publicly available datasets
- A description of any restrictions on data availability
- For clinical datasets or third party data, please ensure that the statement adheres to our policy

Datasets generated in the study:
The raw fcs files are available on community.cytobank.org, experiment numbers 116506, 116507 :
Publicly available datasets that were used through the study:
ImmGen Microarray Phase 1 - GSE15907
Human Protein Atlas - https://www.proteinatlas.org/about/download
Conservation scores from UCSC Genome Browser :
 https://genome.ucsc.edu/cgi-bin/hgTrackUi?db=mm10&g=cons60way
https://genome.ucsc.edu/cgi-bin/hgTrackUi?db=hg19&g=cons100way

## Research involving human participants, their data, or biological material

Policy information about studies with human participants or human data. See also policy information about sex, gender (identity/presentation), and sexual orientation and race, ethnicity and racism.

| Reporting on sex and gender | N/A |
| Reporting on race, ethnicity, or other socially relevant groupings | N/A |
| Population characteristics | N/A |
| Recruitment | N/A |
| Ethics oversight | N/A |

Note that full information on the approval of the study protocol must also be provided in the manuscript.

# Field-specific reporting

Please select the one below that is the best fit for your research. If you are not sure, read the appropriate sections before making your selection.

☒ Life sciences    ☐ Behavioural & social sciences    ☐ Ecological, evolutionary & environmental sciences

For a reference copy of the document with all sections, see nature.com/documents/nr-reporting-summary-flat.pdf

# Life sciences study design

All studies must disclose on these points even when the disclosure is negative.

| Sample size | No statistical calculations were done to predefine mice sample size. We have collected all the available CC strains at the UNC facility as of 2016. The CC mice were collected in duplicates to allow discovery of loci with smaller effect size. |
| Data exclusions | Exclusions were made based on data quality. In particular, low-viability bone marrow samples (by Countess and tripan staining) were excluded. |
| Replication | The CC mice were measured in duplicates, CC founder strains were measured in triplicates. We excluded replicates in case of the low cell viability as described in "Data exclusions". |
| Randomization | The samples were not randomized, as there were no difference in terms of treatment/condition between the mice. |
| Blinding | Experiments were not blinded, as there were no difference in terms of treatment/condition between the mice. |

# Reporting for specific materials, systems and methods

We require information from authors about some types of materials, experimental systems and methods used in many studies. Here, indicate whether each material, system or method listed is relevant to your study. If you are not sure if a list item applies to your research, read the appropriate section before selecting a response.

## Materials & experimental systems

| n/a | Involved in the study |
|-----|----------------------|
| ☐ | ☒ Antibodies |
| ☒ | ☐ Eukaryotic cell lines |
| ☒ | ☐ Palaeontology and archaeology |
| ☐ | ☒ Animals and other organisms |
| ☒ | ☐ Clinical data |
| ☒ | ☐ Dual use research of concern |
| ☒ | ☐ Plants |

## Methods

| n/a | Involved in the study |
|-----|----------------------|
| ☒ | ☐ ChIP-seq |
| ☐ | ☒ Flow cytometry |
| ☒ | ☐ MRI-based neuroimaging |

# Antibodies

| | |
|---|---|
| Antibodies used | All antibodies used in this study are listed in Extended Data Table 2. |
| Validation | Validation of all primary antibodies could be found on the manufacturer's website, along with relevant citations, and antibody profile :<br>Purified anti-mouse TCR β : https://www.biolegend.com/en-us/products/purified-anti-mouse-tcr-beta-chain-antibody-274<br>Purified anti-mouse CD117 : https://www.biolegend.com/en-us/products/purified-anti-mouse-cd117-c-kit-antibody-77<br>Purified anti-mouse CD49b : https://www.biolegend.com/en-us/products/purified-anti-mouse-cd49b-pan-nk-cells-antibody-235<br>Purified anti-mouse CD19 : https://www.biolegend.com/en-us/products/purified-anti-mouse-cd19-antibody-1532<br>Purified anti-mouse CD45 :  https://www.biolegend.com/en-us/products/purified-anti-mouse-cd45-antibody-102<br>Purified anti-mouse CD4 :  https://www.biolegend.com/en-us/products/purified-anti-mouse-cd4-antibody-484<br>Purified anti-mouse Ly-6G  : https://www.biolegend.com/en-us/products/purified-anti-mouse-ly-6g-antibody-4767<br>Purified anti-mouse Ly-6C  : https://www.biolegend.com/en-us/products/purified-anti-mouse-ly-6c-antibody-4894<br>Purified anti-mouse CD8a :https://www.biolegend.com/en-us/products/purified-anti-mouse-cd8a-antibody-157<br>Purified anti-mouse CD115 (CSF-1R) : https://www.biolegend.com/en-us/products/purified-anti-mouse-cd115-csf-1r-antibody-6214<br>Purified anti-mouse CD43 : https://www.biolegend.com/en-us/products/purified-anti-mouse-cd43-antibody-7589<br>Purified anti-mouse Ly-6A/E (Sca-1) : https://www.biolegend.com/en-us/products/purified-anti-mouse-ly-6a-e-sca-1-antibody-230<br>Purified anti-mouse I-A/I-E :  https://www.biolegend.com/en-us/products/purified-anti-mouse-i-a-i-e-antibody-368<br>Purified anti-mouse/human CD11b  : https://www.biolegend.com/en-us/products/purified-anti-mouse-human-cd11b-antibody-351<br><br>Primary conjugates of mass cytometry antibodies were prepared using the Maxpar antibody conjugation kit (Fluidigm Inc.) according to the manufacturer protocol (PRD002 Fluidigm Inc.) and optimal concentration was determined by titration according to the manufacturer protocol. |

# Animals and other research organisms

Policy information about studies involving animals; ARRIVE guidelines recommended for reporting animal research, and Sex and Gender in Research

| | |
|---|---|
| Laboratory animals | The Collaborative Cross founder strain of the laboratory strains A/J, C57BL/6J, 129S1Sv/ImJ, NOD/ShiLtJ, NZO/H1LtJ, and wild-derived strains CAST/EiJ, PWK/PhJ, and WSB/EiJ were purchased from The Jackson Laboratory, delivered to the Systems Genetics Core Facility at The University of North Carolina, and sacrificed. Altogether, we profiled 23 founder mice aged 6- 8 weeks. The Collaborative Cross recombinant mice were both purchased from, handled, bred and sacrificed by the Systems Genetics Core Facility at The University of North Carolina. Altogether, we profiled 129 Collaborative Cross mice aged 8-14 weeks.<br>Mice handaling: mice were group-housed in GM500 Green Line individually ventilated caging (Tecniplast, Buguggiate, Italy) with 70 air exchanges per hour. The room was maintained between 70 and 74°F with 30 -70 % humidity and a 12-h light cycle. |
| Wild animals | The study did not include wild animals |
| Reporting on sex | All the mice were male |
| Field-collected samples | The study did not include samples collected from the field |
| Ethics oversight | All procedures involving animals were performed according to the Guide for the Care and Use of Laboratory Animals with prior approval by the Institutional Animal Care and Use Committee within the Association for Assessment and Accreditation of Laboratory Animal Care-accredited program at the UNC at Chapel Hill (Animal Welfare Assurance Number: A-3410-01) |

Note that full information on the approval of the study protocol must also be provided in the manuscript.

## Plants

Seed stocks: N/A

Novel plant genotypes: N/A

Authentication: N/A

## Flow Cytometry

### Plots

Confirm that:

☒ The axis labels state the marker and fluorochrome used (e.g. CD4-FITC).

☒ The axis scales are clearly visible. Include numbers along axes only for bottom left plot of group (a 'group' is an analysis of identical markers).

☒ All plots are contour plots with outliers or pseudocolor plots.

☒ A numerical value for number of cells or percentage (with statistics) is provided.

### Methodology

Sample preparation: Cells from each sample were washed twice with Cell Staining Medium (Maxpar) and a total of three million cells were used for extracellular staining. Cells were resuspended in 500 µl containing 1:2000 rhodium DNA intercalator (Fluidigm) for 20 min of live/dead cells staining. Samples were washed with Cell Staining Medium and resuspended in a total of 100 µl metal tagged antibody mix for 1 hour, for cell surface marker staining. Cells were then fixed with 1.6% PFA (Sigma-Aldrich) in a total volume of 200 µl and stored at 4°C. Cells were centrifuged, PFA removed and iridium DNA intercalator staining was performed for 20 min at 1:2000 dilution in 500 µl volume to differentiate cells from debris. Finally, fixed samples were washed 3 times with deionised water immediately prior to data acquisition.

Instrument: CyTOF1 machine (DVS Sciences)

Software: CyTOF 1 acquisition software (v1)

Cell population abundance: Cell population abundances were defined by manual gating. The resulting frequencies are available in Extended Data Tables 4 and 6.

Gating strategy: Detailed gating strategy is shown in Supplementary Figure 1.

☒ Tick this box to confirm that a figure exemplifying the gating strategy is provided in the Supplementary Information.

