## [Peer Review file · Nature]

Manuscript Title: Interactions between immune cell types facilitate the evolution of immune traits

Reviewer Comments & Author Rebuttals

Reviewer Reports on the Initial Version:

Referees' comments:

Referee #1 (Remarks to the Author):

The manuscript presents novel and important findings about immune system genetic architecture and evolvability. The study reveals considerable variation in immune profiles in bone marrow in a set of genetically diverse mouse strains. This variation is shown to be polygenic, and detailed analysis indicates that causal variants can influence the frequency of cells in which they are not expressed. These findings suggest that the immune system's organization, which relies on interactions of multiple cell types, improves the immune system's mutational robustness and, hence, its evolvability.

The argumentation is sound and fully supported by the statistical analysis of the data. The introduction clearly outlines how the study addresses the big question of immune systems' evolvability and convincingly argues that the Collaborative Cross panel is a relevant tool for studying the genetic architecture of immune traits.

My only reservation here is that the abstract states that "organization into interacting modules (...)" is "a key determinant" of evolvability - I think this statement would need further support (perhaps a more extensive discussion of the literature) in addition to the findings presented in the manuscript which are limited to the study of only one phenotype, the frequency of different cell types in the bone marrow.

Overall, the presentation is clear, with a few exceptions I discuss below, suggesting possible improvements.

My primary reservation about how the discussion of the results is laid out concerns the definition of the "cyto-cis" and "cyto-trans" genes. The presentation should emphasize that as long as these labels refer to two distinct sets of genes, as defined in the paper, both can act in trans.

Consequently, I find phrasing "exclusively cyto-cis" (as in reference to Figure 4a, page 9) confusing. The fact that "cyto-cis" genes can also have trans effects also seems to be important in interpreting Figures 4c and 4d (bottom of page 9), and this fact should be highlighted there.

Other comments and suggested improvements:

1. Please add page and line numbers; it would make reviewing much easier.
2. Page 2, paragraph 3: suggest removing "thus" since the implication is unclear.
3. Page 2, paragraph 3: suggest adding "cell types' frequencies" to clarify the statement about the findings in reference 27.
4. Figure 2a: what is the percentage of loci that did not pass the validation by the second cohort?
5. Figure 2d: what fraction of the genes contributing to the three categories are shared?

6. Page 5, paragraph 2, "Thus, akin to mRNA or protein abundance, cell subset abundance is largely determined by genetic variants of turnover genes.": explain and/or provide a reference
7. Figure 3: suggest relabeling the c and d subplots for consistency
8. Figure 3b: Since the two subplots share one axis, how about plotting cell types as y and expression/LOD score as x to aid the comparison?
9. Figure 3c: Current organization suggests a complex pattern that is, in fact, quite simple. The cell types in expressed and associated categories could instead be sorted in concordance to highlight the simplicity of the pattern and emphasize exceptions (i.e., Stem-Stem, Pro-B--Pro-B, Late B--Late B, etc.).
10. Figure 3d: how many cis and trans associations are the same across cell types? Suggest adding a supplementary heatmap to answer this question.
11. Figure 4d: For completeness, the distributions of a.a. conservation reported here should be juxtaposed with the distribution of cyto-cis and cyto-trans genes affecting a single cell type. For interpretation, it would be informative to show this distribution for all genes and all genes expressed in immune cell types, as well as with the group of turnover-associated cells referenced before.
13. Page 11, paragraph 2: suggests pointing out that the immune systems' modularity is exceptional
14. Page 14, "Signal propagation": justify the choice of the 40% threshold.

Referee #2 (Remarks to the Author):

The manuscript 2023-07-13490, entitled "Interactions between immune cell types facilitate the evolution of immune traits", by Dubovik et al. reports evidence that genes affecting immune cells in which they are not expressed are under weaker selective constraints relative to genes affecting immune cells in which they are expressed. The authors interpret this result as evidence that cell-cell interactions have played an important role in the recent evolvability of the immune system.

The manuscript is very well-written and the analyses are generally straight-forward. The conclusions are original and potentially interesting, although I believe that several additional analyses are needed to confirm and strengthen the authors' conclusions. Furthermore, several limitations should be discussed more explicitly.

Major comments:

1. The authors consider that evidence for weaker negative selection is evidence for greater evolvability, which is not strictly equivalent. As defined by Payne and Wagner (2019), "evolvability is the ability of a biological system to produce phenotypic variation that is both heritable and adaptive". The concept of evolvability thus implies that biological systems evolve to become more genetically variable, and that the resulting genetic variation will eventually be adaptive. In this study, the authors conclude that non-cell-intrinsic genes are more "evolvable" because they show lower evolutionary conservation (i.e., more divergence across species), compared to cell-intrinsic genes. However, they do not show that variation in non-cell-intrinsic genes is more likely to be adaptive. Indeed, genes under weak negative selection are (i) less important for survival or reproduction, relative to genes under strong negative selection, (ii) are expected to evolve under more neutral evolution and (iii) are more likely to be lost. However, this does not imply that they are more likely to be adaptive. To support the conclusion that cell-cell interactions "facilitate the evolution of

immune traits”, the authors should instead show that non-cell-intrinsic genes are enriched in genes under positive selection, compared to cell-intrinsic genes. This can be achieved using interspecies or intraspecies tests in wild mice (Lawal et al., BMC Biology 2021; Booker et al., bioRxiv 2021).

2. It is unclear why the authors have focused on protein-coding variants. Many of the associations with blood cell traits in humans are in intergenic or intronic regions and colocalize with eQTLs (Chen et al., Cell 2020; Akbari et al., Nat Commun 2023), as most other traits (Maurano et al., Science 2012). I suggest that the authors extend their analyses to non-coding regions. Non-coding variation can be associated to genes by eQTL databases. Of note, associations with coding variation are more prone to false positives, as these variants can affect antibody affinity rather than cell count.

3. Crossings between inbred mice are a very useful system to detect genotype-phenotype associations. Nevertheless, this is an artificial system in which genotypic and phenotypic variation is affected by artificial selection and inbreeding depression (Philip et al., Genome Res 2011). I thus strongly suggest that the authors replicate their findings in *Homo sapiens*, to strengthen their conclusions. The authors have arguably not investigated human genes because of the low resolution of the study by Vuckovic et al., Cell 2020 (5 cell types are investigated, vs. 9 in this study). However, several other studies have identified hundreds of human genes affecting blood cell parameters measured at higher resolution than the current study (see Orrù et al., Nat Genet 2020; Chen et al., Cell 2020; Akbari et al., Nat Commun 2023), providing sufficient power to test their hypothesis.

4. The authors assume that genes that affect the frequencies of those cell types in which they are not expressed are genes involved in cell-cell interactions, but cyto-trans genes are only weakly enriched for extracellular localization. Actually, gene expression can vary widely during hematopoiesis, so that genes that are not expressed in differentiated cell types may have been expressed in these cell types during earlier developmental stages. Can the authors leverage databases of gene expression during hematopoiesis (e.g., Choi et al., Nucleic Acids Res 2019) to reinforce their conclusions?

5. It is unsurprising that “associated genes are largely involved in quantitative homeostasis through cell-intrinsic functions of proliferation, migration and cell death”, as they were pre-selected to be expressed in immune cells. Can the authors check whether associated genes are enriched in such ontologies, compare to the 15,470 tested loci? What is the background gene set when testing enrichment with Ingenuity Pathways (e.g., LPS response)?

Minor comments

- The authors have considered as candidate genes those genes that carry “a variant in this gene expected to severely alter the protein or the transcript” in at least one founder strain. Can the authors verify whether these genes are under weaker negative selection than all other genes? This may affect the comparison between non-cell-intrinsic genes and cell-intrinsic genes, if their proportions are different in the strongly constrained genes that are not tested.
- Important details are lacking for the validation step. Is the direction of effects (observed in the discovery cohort) replicated in the validation cohort? What is the proportion of replicated variants? Can the authors clarify if the validation cohort includes mice that were already present the discovery

cohort (Extended Data Table 1)? If so, the analysis should be redone, as a replication cohort should be independent from the discovery cohort.

- The analysis named “Signal propagation” is unclear. How did the authors use the associated clusters? Did they include clusters when performing enrichment tests? The way associated clusters are defined (i.e., median LOD score of genes in the cluster for the given cell type is at least 40% of the highest median calculated for that cluster) seems arbitrary. Can the authors justify the approach?
- Please avoid terms that are uncommon and/or not properly defined. What do the authors mean by “quantitative homeostasis”, “useful variation”, “gain advantage in evolvability”?
- Please replace “fewer purifying constraints” by “weaker negative selection”, which is more conventional.
- First paragraph of the introduction: To my knowledge, there is no evidence that “more complex” organisms evolve at the slower evolutionary rate than “less complex” organisms. Please clarify.
- Please rephrase “This suggests that the observed variation in immune profiles is likely to possess adaptive value in face of bacterial threats.” This is an overstatement; a gene enrichment is not suggestive of any “adaptive value”.
- The statement “while the genetic determinants of inner cell life have been fixed early in the evolution to ensure robust functioning of the individual immune cell types” is highly speculative and questionable. Note that fixation is rare in genes under strong negative selection. Similarly, the statement “the interaction between different immune cell types provides a flexible phenotypic space where mutations can produce selectable variation without much detriment.” Is highly speculative. The authors provide no evidence that such variation will be adaptive.
- Several figure panels are not described or discussed in the text (Figs. 2b, 3c, etc.).
- Why does the LODscore significance threshold vary from one cell type to another? Sample size? Please clarify in the Figure 3b legend.
- Fig. 4b does not show age estimates but amino-acid conservation score, so it cannot support the statement that “genes associated with the frequency of multiple cell types are evolutionary younger”.
- Can the authors briefly discuss how variants on the MegaMUGA SNPchip were ascertained and if this ascertainment can bias their results?
- According to EU legislation (2010/63/EU), the term “euthanasia” should be avoided, as it implies that mice were suffering at the time of the killing, and should be replaced by “killing” (<https://www.nc3rs.org.uk/3rs-resources/euthanasia>).
- Can the authors illustrate with a Suppl. Figure how the cell count data changes after within-sample, over-time normalization?

Referee #3 (Remarks to the Author):

The article aims to demonstrate that genetic factors influencing the frequency of immune cell types are expressed within the corresponding cell types, thus exhibiting cell-intrinsic functions (cyto-cis genes). Additionally, the study reveals that certain genes impacting the frequency of a particular cell type are not expressed within these cells but rather in other cells within the immune system, indicating a function in trans (cyto-trans genes).

I have several questions/ concerns listed in the following paragraph

Abstract

Lane 4 The authors wrote:

"The immune system is one of the fastest-evolving components of mammalian genomes. They cite Shultz AJ and Sackton TB immune genes serve as hotspots for shared positive selection across birds and mammals Elife 2019.

I am curious about the gene overlap between The two studies .

In Lane 16, (and also in the discussion session) the authors state, 'Our findings suggest that interactions between different components of the immune system provide a phenotypic space where mutations can produce selectable variation without significant detriment, offering a potential resolution to the robustness-evolvability conundrum within the context of the immune system.'

They reference the work of Wagner and colleagues.

However, in a more recent paper, the same research group contradicts this statement. They argue that robustness actually increases evolvability (Zheng J, Guo N, Wagner A. 'Selection enhances protein evolvability by increasing mutational robustness and foldability.' Science. 2020 PMID: 33273072)

Main text

Lane 2 the authors wrote : While in simple systems, random mutations are likely to produce beneficial phenotypes at a non-negligible frequency.

The authors need to provide reference here.

4) Lane 8: the authors wrote The immune system is a complex system intricately involved in the battle against pathogens, making it a prime target for the process of natural selection.

It's worth noting that the immune system also plays a role in maintaining homeostasis.

Could the author clarify if the cell type corresponds to the module? This is what I understand from the text.

5) Lane 20

"The authors cite many articles looking at the genetic determinants of immune profiles (DOI: 10.1016/j.cell.2020.08.008). I think it is important to investigate the overlap between the candidate genes found in their study and the studies conducted by other investigators.

6) Page 4 variation in immune profile is polygenic

""The author wrote, 'We consider only genes that are expressed in at least one immune cell type.' However, in their study, they demonstrate that genes expressed in trans affect the frequency of different cell types. I believe that by restricting their consideration to genes expressed in at least one immune cell type, they may overlook some potential gene candidates (that are not expressed in immune cell type) Maybe I miss something : please clarify

7) It would also helpful to have the ratio of coding versus non coding variation that impact the phenotype and compare their data with DOI: 10.1016/j.cell.2020.08.008

8) Generally, do cyto-trans genes code for cytokines or other soluble elements? In other words, are some of these genes already known to function in trans?"

More specifically, is Bex1 known to act in trans?

9) Page 9: Cyto-trans genes and evolution

I believe this subtitle is somewhat misleading. The authors demonstrate that cyto-trans gene evolution is weakly constrained, but they do not necessarily show that these genes actively facilitate evolution.

General remarks to all reviewers:

We thank the reviewers for their careful consideration of our work and for the constructive suggestions for improvement. We have now implemented most of the suggestions, expanded the analysis and present a revised manuscript.

Most notably, we have now validated our main finding in another species, humans, via previously published data coming mostly from the UK biobank. We show that also for the genes associated with the variation in immune cell type frequencies in human blood, genes that influence cell types in which they are not expressed (i.e. associated in *cyto-trans*) are evolutionarily less conserved. This expansion of our work, using data published previously and from another species, effectively corroborates our main findings.

We have also taken advantage of the reviewers' expert advice to solidify the technical aspects of our experimental work. Using these more stringent criteria has resulted in fewer genes found to be associated with immune profiles in the mouse bone marrow. However, this smaller set of associated genes shows an even stronger signal in terms of differences in sequence conservation between genes associated in *cyto-trans* vs. *cyto-cis*! We thank all the reviewers for helping to bring forward the signal in our data.

We present the respective point-by-point responses below.

Referee #1:

The manuscript presents novel and important findings about immune system genetic architecture and evolvability. The study reveals considerable variation in immune profiles in bone marrow in a set of genetically diverse mouse strains. This variation is shown to be polygenic, and detailed analysis indicates that causal variants can influence the frequency of cells in which they are not expressed. These findings suggest that the immune system's organization, which relies on interactions of multiple cell types, improves the immune system's mutational robustness and, hence, its evolvability.

The argumentation is sound and fully supported by the statistical analysis of the data. The introduction clearly outlines how the study addresses the big question of immune systems' evolvability and convincingly argues that the Collaborative Cross panel is a relevant tool for studying the genetic architecture of immune traits.

My only reservation here is that the abstract states that "organization into interacting modules (...)" is "a key determinant" of evolvability - I think this statement would need

further support (perhaps a more extensive discussion of the literature) in addition to the findings presented in the manuscript which are limited to the study of only one phenotype, the frequency of different cell types in the bone marrow.

We thank the reviewer for their careful reading and for the encouraging words. We agree that our findings show that the organisation into interacting modules is an important determinant of evolvability, but that without further research we cannot be sure it is the key determinant of its evolvability. We have replaced the wording 'key determinant' with 'important determinant' (line 21).

Overall, the presentation is clear, with a few exceptions I discuss below, suggesting possible improvements.

My primary reservation about how the discussion of the results is laid out concerns the definition of the "cyto-cis" and "cyto-trans" genes. The presentation should emphasize that as long as these labels refer to two distinct sets of genes, as defined in the paper, both can act in trans. Consequently, I find phrasing "exclusively cyto-cis" (as in reference to Figure 4a, page 9) confusing. The fact that "cyto-cis" genes can also have trans effects also seems to be important in interpreting Figures 4c and 4d (bottom of page 9), and this fact should be highlighted there.

We regret the unfortunate wording. Indeed, genes that act as *cyto-cis* with respect to one trait can have *cyto-trans* effects with respect to other traits. The phrasing of 'exclusively *cyto-cis*' was merely intended to express that no *cyto-trans* effect was observed in our study, but we did not intend to say that these genes cannot have *cyto-trans* effects for other phenotypes not assayed in this study. To better reflect this meaning, we have amended the wording to: 'genes for which we found a *cyto-trans* association in our study' (line 227) and 'Genes are grouped by whether they were found associated with immune cell frequencies only in *cyto-cis* or also in *cyto-trans*' (Caption to **Fig. 4**).

Other comments and suggested improvements:

1. Please add page and line numbers; it would make reviewing much easier.

We apologise for having omitted this simple formatting aid. Page and line numbers are now included.

2. Page 2, paragraph 3: suggest removing "thus" since the implication is unclear.

Removed (line 55).

3. Page 2, paragraph 3: suggest adding "cell types' frequencies" to clarify the statement about the findings in reference 27.

The statement was amended for clarity in line with the suggestion (line 61).

4. Figure 2a: what is the percentage of loci that did not pass the validation by the second cohort?

Following reviewers' comments we have now applied more stringent criteria, most notably the inclusion of the criterion of validating the directionality of effects in the validation cohort. The percentage has now increased and stands at 85%. Overall, this resulted in fewer associated genes, however, the main finding that genes associated in cyto-trans are evolutionary less conserved, manifests yet more clearly compared to our original submission (**Fig. 4a**).

Fig. 4a: Empirical cumulative distribution functions for amino acid conservation scores of genes grouped by whether they were found associated with immune cell frequencies only in *cyto-cis* or also in *cyto-trans*. *P*-value for one-sided Kolmogorov-Smirnov test between these two groups is stated. Cumulative distributions of evolutionary conservation for immune genes considered in this study (*Methods*) and for the non-immune genes in the mouse genome are shown for comparison in grey.

5. Figure 2d: what fraction of the genes contributing to the three categories are shared?

The sharing of genes between the categories is illustrated in the Venn diagram below. The majority (60%) of genes belong only to a single of the three categories.

R2R Figure 1: Venn diagram of mouse genes associated with the immune profiles in the bone marrow belonging to the functional groups of cellular death, proliferation and cell migration.

6. Page 5, paragraph 2, "Thus, akin to mRNA or protein abundance, cell subset abundance is largely determined by genetic variants of turnover genes.": explain and/or provide a reference

We thank the reviewer for pointing out the lack of clarity. We merely intended to provide a rationalisation of why it is unsurprising that many genes associated with cell type frequencies are involved in cell proliferation, cell death or movement of cells (presumably out of the bone marrow) by likening it to protein or mRNA abundance, which can also be expressed as a function of their turnover rates (synthesis minus degradation). We have tried to clarify this statement in the revised manuscript (lines 145-147): 'Taken together, akin to the control of mRNA and protein abundance, cell subset abundance is also subject to turnover rates.'

7. Figure 3: suggest relabeling the c and d subplots for consistency

We have relabelled the panels.

8. Figure 3b: Since the two subplots share one axis, how about plotting cell types as y and expression/LOD score as x to aid the comparison?

We thank the reviewer for this neat suggestion, which we have implemented.

9. Figure 3c: Current organisation suggests a complex pattern that is, in fact, quite simple. The cell types in expressed and associated categories could instead be sorted in

concordance to highlight the simplicity of the pattern and emphasize exceptions (i.e., Stem-Stem, Pro-B--Pro-B, Late B--Late B, etc.).

We have now sorted the labels in concordance as suggested. However, thanks to this suggestion, we also realised that the original plot could be improved in clarity, as graphically the most focus was placed on *cyto-cis* associations. We have now amended the Circos plot to only show the *cyto-trans* associations, highlighting that there is no simple pattern of *cyto-trans* associations across the cell types.

10. Figure 3d: how many cis and trans associations are the same across cell types? Suggest adding a supplementary heatmap to answer this question.

There are 14 genes for which we found both *cyto-cis* and *cyto-trans* associations. To present this in more detail, we have now added a heatmap addressing this question as a panel in one of the extended data figures (**Extended Data Fig. 4a**).

Extended Data Fig. 4a: Circular heatmap of associations between genes and immune cell types, classified according to expression in the respective cell type. Genes are clustered according to their association profile across cell types. Cell types are indicated on each ring (CD4⁺ – CD4⁺ T cells, CD8⁺ – CD8⁺ T cells, GN – Granulocytes, MO – monocytes).

11. Figure 4d: For completeness, the distributions of a.a. conservation reported here should be juxtaposed with the distribution of cyto-cis and cyto-trans genes affecting a single cell type. For interpretation, it would be informative to show this distribution for all genes and all genes expressed in immune cell types, as well as with the group of turnover-associated cells referenced before.

We have now added the cumulative distribution of amino acid conservation scores for genes affecting a single-cell type, classified by *cyto-cis* or *cyto-trans*, as **Extended Data Fig. 4c**; there is no significant difference between the two distributions. We have also included the distribution for all immune genes and all the other genes as part of **Fig. 4a**, so that the distributions for associated genes, *cyto-trans* and *cyto-cis*, can be directly juxtaposed with these.

Fig. 4a: Empirical cumulative distribution functions for amino acid conservation scores of genes grouped by whether they were found associated with immune cell frequencies only in *cyto-cis* or also in *cyto-trans*. *P*-value for one-sided Kolmogorov-Smirnov test between these two groups is stated. Cumulative distributions of evolutionary conservation for immune genes considered in this study (*Methods*) and for the non-immune genes in the mouse genome are shown for comparison in grey.

13. Page 11, paragraph 2: suggests pointing out that the immune systems' modularity is exceptional

We agree that due to the strong pressure on the immune system to evolve, the immune system is poised to manifest the modularity exceptionally well. We have now included a statement to that end in the *Discussion* (lines 313-319):

'While evolvability through modularity is not expected to be limited to the immune system, due to the high need for its rapid evolution, the immune system is poised to manifest modularity exceptionally strongly. Further research into immune system evolvability thus could not only enlighten the design principles behind immune responses, but also contribute to biomimetic solutions e.g. in the system-of-systems approach to engineering, which is similarly based on interactions between functional units⁵⁹.'

14. Page 14, "Signal propagation": justify the choice of the 40% threshold.

The 40% threshold was chosen empirically based on the distribution of the relative association scores, which has a trough at 0.4 in between two peaks. This explanation is now included in the *Methods* and the plot of the distribution is now included as **Extended Data Figure 3b**.

Extended Data Figure 3b: Distribution of relative median LOD scores for gene clusters during signal propagation. Genes found significantly associated to at least immune cell type were clustered based on their pattern of association across all the assayed cell types. For each cluster, median LOD score for each cell type was calculated and then divided by the maximum value of this quantity across all cell types. The distribution of the resulting values is plotted, with two clear peaks – all gene cluster-cell type pairs with median LOD score above 0.4 were considered associated.

Referee #2:

The manuscript 2023-07-13490, entitled “Interactions between immune cell types facilitate the evolution of immune traits”, by Dubovik et al. reports evidence that genes affecting immune cells in which they are not expressed are under weaker selective constraints relative to genes affecting immune cells in which they are expressed. The authors interpret this result as evidence that cell-cell interactions have played an important role in the recent evolvability of the immune system.

The manuscript is very well-written and the analyses are generally straight-forward. The conclusions are original and potentially interesting, although I believe that several additional analyses are needed to confirm and strengthen the authors’ conclusions. Furthermore, several limitations should be discussed more explicitly.

We thank the reviewer for carefully reading our manuscript and for appreciating the novelty of our conclusions. We especially thank the reviewer for suggesting to replicate our findings in the human data, which we have enthusiastically done to a great avail. We address all the potential limitations of our findings below.

Major comments:

1. The authors consider that evidence for weaker negative selection is evidence for greater evolvability, which is not strictly equivalent. As defined by Payne and Wagner (2019), “evolvability is the ability of a biological system to produce phenotypic variation that is both heritable and adaptive”. The concept of evolvability thus implies that biological systems evolve to become more genetically variable, and that the resulting genetic variation will eventually be adaptive. In this study, the authors conclude that non-cell-intrinsic genes are more “evolvable” because they show lower evolutionary conservation (i.e., more divergence across species), compared to cell-intrinsic genes. However, they do not show that variation in non-cell-intrinsic genes is more likely to be adaptive. Indeed, genes under weak negative selection are (i) less important for survival or reproduction, relative to genes under strong negative selection, (ii) are expected to evolve under more neutral evolution and (iii) are more likely to be lost. However, this does not imply that they are more likely to be adaptive. To support the conclusion that cell-cell interactions “facilitate the evolution of immune traits”, the authors should instead show that non-cell-intrinsic genes are enriched in genes under positive selection, compared to cell-intrinsic genes. This can be achieved using interspecies or intraspecies tests in wild mice (Lawal et al., BMC Biology 2021; Booker et al., bioRxiv 2021).

We thank the reviewer for the opportunity to clarify the scope of our argument. As the authors quoted by the reviewer have shown in their earlier work on transcription factor binding sites (Payne & Wagner, *Science* 2014, PMID: 24558158), variation does not need to be immediately adaptive in order to support evolvability. Instead, Payne & Wagner found that the more phenotypically robust the particular component is in the face of genetic variation, the higher evolvability it possesses. Phenotypically near-neutral variation (i.e. non-detrimental and non-adaptive at the same time) in a functionally important component is in itself sufficient to support evolvability, as some of the near-neutral variants may eventually become adaptive in a new environment.

Taking this argument to its logical consequence, should any of the individual genes providing evolvability through robust variation be subjected to strong positive selection, it would cease to provide evolvability, as it would cease to provide phenotypically near-neutral variation.

In our work, we have found genetic determinants of variation in immune profiles. Immune profiles have been found to be an important determinant for a number of survival-related traits, as we have detailed in our introduction. Thus, genes found linked to the variation in immune profiles are causing variation in a functionally important trait. A subset of these genes, those that are not expressed in the cell type the frequency of which they influence (*cyto-trans* genes), provide from an evolutionary perspective, a larger extent of near-neutral variation than the rest of the associated genes (*cyto-cis* genes). We thus concluded that the *cyto-trans* genes are supporting the evolvability of the immune system by providing more of the near-neutral variation. This is true irrespective of whether, in the course of evolution, the near-neutral variation in these genes has been acted upon and subjected to positive selection, or if such

positive selection is yet to take place. As a consequence, absence or presence of positive selection in the evolutionary record is insufficient both to refute and to support that the *cyto-trans* genes enhance the immune system's evolvability.

Nevertheless, we conducted the analysis suggested by the reviewer. We found a relatively small number of genes associated with immune profiles to be contained in candidate selective sweep loci in *M. m. musculus* identified by Lawal et al. (BMC Biology 2021). Specifically, these regions contained 6 out of 260 cyto-cis genes: *Col4a1*, *Ppip5k2*, *Ptprm*, *Rap2b*, *Txn11*, *Upf2*; and 3 out of 21 cyto-trans genes: *Mme*, *Ranbp17*, *Serpinb2*. This corresponds to a significant enrichment of positively selected regions for cyto-trans genes (one-sided Fisher exact test $P = 0.023$). In any case, due to reasons stated above, we do not consider this to be supporting evidence for our main claim.

We have now amended the manuscript to clearly reflect the scope of our claim. We have significantly rephrased the second paragraph in the Discussion to clearly address that it is the absence of negative selection rather than the presence of positive selection that is crucial for evolvability (Lines 291-301):

'While variation with strong positive or negative fitness effects have appreciated roles in the evolution of biological species, the importance of near-neutral variation has only lately emerged as an important consideration for the capacity of biological systems to evolve. Recent work on transcription factor and protein structure evolution^{2,7,58} has found that increased capacity for near-neutral variation, i.e. increased mutational robustness, paradoxically facilitates evolution by supporting genetic diversity. This has solved the apparent conflict between evolvability and robustness⁸ – the seemingly contradictory requirement that phenotypes be significantly altered by genetic changes to allow selection of fitter phenotypes, but not be altered too much to endure frequent mutations without harm. Our results offer an appealing extension of this evolvability principle in the context of the immune system – an enhancement of the near-neutral genetic diversity is achieved through modularity.'

2. It is unclear why the authors have focused on protein-coding variants. Many of the associations with blood cell traits in humans are in intergenic or intronic regions and colocalize with eQTLs (Chen et al., Cell 2020; Akbari et al., Nat Commun 2023), as most other traits (Maurano et al., Science 2012). I suggest that the authors extend their analyses to non-coding regions. Non-coding variation can be associated to genes by eQTL databases. Of note, associations with coding variation are more prone to false positives, as these variants can affect antibody affinity rather than cell count.

We concur with the reviewer that only a small fraction of associated sites are expected to be found in the regions that we selected for our analysis (e.g. according to a meta-analysis by Maurano et al, 2012, referenced by the reviewer, only about 5% of all SNPs associated to various traits in a total of 920 GWAS studies are found in coding regions). However, in this study we prioritised interpretability over comprehensiveness. In order to assess the evolvability of immune profiles in terms of the competing limitations by a specific protein function (in this case,

its expression or non-expression in the associated cell-type) we needed to clearly link the associated locus to a specific gene. This is highly problematic for variants in intergenic regions, unless, as the reviewer is suggesting, an eQTL mapping is available. However, eQTLs are dependent on species and even particular cell types (cf. Yazar et al, *Science* 2022, PMID: 35389779), and we are not aware of an eQTL resource for mouse bone marrow stratified by cell type. This would further complicate the interpretation of these associated variants. This is an important consideration given that Borsari et al. (*Genome Research* 2021, PMID: 34290042) found that intronic enhancers active in the blood actually control the gene containing the intron less frequently than some other gene.

We agree though that focus on the coding sequence only might be unnecessarily limiting in scope. We thus currently include all variants in the exons. This excludes intronic and intergenic variants for the reasons stated above, but includes variants in non-coding regions such as 5'UTRs, which can be easily interpreted. We consider this focus on the exonic sequence a strength rather than a weakness of our argument, since these variants can be linked to the containing gene with reasonable confidence.

In our experiments, we indeed observed that variation in the coding sequence had an effect on the antibody affinity. To prevent false positives in this regard, we conducted manual rather than automated gating to differentiate cells that are positive vs. negative for a given surface protein. In this way, the threshold is customised for each strain and cell type, ensuring that the exact antibody affinity plays no role in determining the cell identity and consequently the frequency of a particular cell type. In cases where the antibody binding has been lost altogether, i.e. no cells positive for the given antigen could be detected, these strains have been excluded completely from the analysis as we detailed in the *Methods*. In this way, we believe we have prevented the occurrence of false positives due to alterations in antibody binding affinities. In any case, out of 14 phenotypic markers used in this study, we found a single one of them associated (*Ly6c*) to immune profiles, so at worst, one out of 271 genes would be a false positive due to the variation in antibody affinity.

We have now amended the manuscript to better justify the rationale for preselecting exonic variants for the association study (lines 113-118):

'To increase the interpretability of our association study, we only considered genes with a potential function in the immune system as identified previously by the ImmGen consortium (a broad set of 7,965 genes)^{46,47} and such that at least one founder strain contains an exonic variant in this gene, for a total of 6,902 genes represented by 15,458 loci (Methods). The exclusion of intronic and intergenic variants was driven by our goal to associate the variants to specific genes.⁴⁸'

3. Crossings between inbred mice are a very useful system to detect genotype-phenotype associations. Nevertheless, this is an artificial system in which genotypic and phenotypic variation is affected by artificial selection and inbreeding depression (Philip et al., *Genome Res* 2011). I thus strongly suggest that the authors replicate their findings in *Homo sapiens*, to strengthen their conclusions. The authors

have arguably not investigated human genes because of the low resolution of the study by Vuckovic et al., Cell 2020 (5 cell types are investigated, vs. 9 in this study). However, several other studies have identified hundreds of human genes affecting blood cell parameters measured at higher resolution than the current study (see Orrù et al., Nat Genet 2020; Chen et al., Cell 2020; Akbari et al., Nat Commun 2023), providing sufficient power to test their hypothesis.

We thank the reviewer for this suggestion – we followed up on it and are delighted to report that we observe a similar phenomenon in the human.

The principle that *cyto-trans* genes promote evolvability has no reason to be specific to mice – indeed, this is a principle that we propose should be true of any sufficiently advanced cell-based immune system. Therefore it indeed is critical to validate these findings in a different species and in a non-artificial setting.

Specifically, we have combined data on blood cell type associations from existing studies in the human (Vuckovic et al., Cell 2020, Chen et al., Cell 2020). In this way, we could assemble a total of 1548 associations with 5 cell types. We have then used The Human Protein Atlas to categorise these associated genes as expressed or non-expressed in the associated cell type similarly as we did for our data in the mouse. Combining this data with the evolutionary record on sequence conservation, we reproduced our main finding (**Fig. 4d**). Thus, we confirmed that what we observed in an artificial system in the mouse is true also for natural variation in a different species. We have now included these results in the manuscript.

Figure 3d: Empirical cumulative distribution functions for amino acid conservation scores of genes found associated with immune cell frequencies in the human blood by previous studies. Genes are grouped by whether they were found associated with immune cell frequencies only in *cyto-cis* or also in *cyto-trans*.

4. The authors assume that genes that affect the frequencies of those cell types in which they are not expressed are genes involved in cell-cell interactions, but *cyto-trans* genes are only weakly enriched for extracellular localization. Actually, gene expression can vary widely during hematopoiesis, so that genes that are not expressed in differentiated cell

types may have been expressed in these cell types during earlier developmental stages. Can the authors leverage databases of gene expression during hematopoiesis (e.g., Choi et al., Nucleic Acids Res 2019) to reinforce their conclusions?

We thank the reviewer for the comment and apologise for the lack of clarity. Of note, in our manuscript we talk about interaction between cell types rather than cells. This semantic distinction is important – as our use of the word interaction is not in the narrow sense of ‘physical interaction between cells’, but rather in its broader meaning which includes e.g. genetic interaction, chemical interaction or indirect interactions. Correspondingly, we discuss the interactions between cell types in the sense of how cells of one cell type influence the count of another cell type. Interaction here is meant in this sense of the word, as we stated in our original manuscript (now lines 186-189).

In this sense, the interaction between a progenitor and the quantity of its differentiated progeny is an interaction between cell types. In fact, genes that exhibit *cyto-trans* interactions are frequently expressed in hematopoietic stem cells. This is now clearly visible in the new version of **Fig. 2d**, where all *cyto-trans*-associations are plotted. This means that indeed a large part of the *cyto-trans* genes had been expressed in the earlier developmental stages of the associated cell type. This in no way diminishes our finding that proteins that do not possess the burden of additional function in the associated cell types (as implied by the fact that they are not even expressed there) are a more potent source of near-neutral variation and hence, support evolvability. This further supports the conclusion that the need for evolvability should be considered as a driving factor behind the organisation of the immune system into many cell types.

Fig 2d: Split Circos plot, depicting expression of genes associated with immune cell frequencies in *cyto-trans*. The colour of connecting lines is determined by the cell type with which the gene is associated; the colour of the rim on the left hand side corresponds to the cell type in which the gene is expressed. For associated genes expressed in multiple cell types, multiple lines are shown. GN – granulocytes.

5. It is unsurprising that “associated genes are largely involved in quantitative homeostasis through cell-intrinsic functions of proliferation, migration and cell death”, as they were pre-selected to be expressed in immune cells. Can the authors check whether associated genes are enriched in such ontologies, compare to the 15,470 tested loci? What is the background gene set when testing enrichment with Ingenuity Pathways (e.g., LPS response)?

We agree with the reviewer that the fact that genes involved in proliferation, cell migration (e.g. out of the bone marrow) and cell death are associated with the relative quantity of specific immune cells in the bone marrow is unsurprising, at least in retrospect. However, this is not due to the preselection, as the functional enrichment of associated genes was done using the pre-selected genes as the background set. Rather, this is likely due to the fact that for any dynamical process, factors influencing the birth and death of any entity are necessarily influencing the balance of the entity. This is what we tried to express with the term ‘quantitative homeostasis’, which we now try to better capture with the term ‘homeostatic balance’ in the revised manuscript (Lines 144-147): ‘...31% of the genes [were] annotated for at least one of the functions of proliferation, cell death, and cellular movement (**Fig. 2d**), i.e. basic determinants of homeostatic balance. Taken together, akin to the control of mRNA and protein abundance, cell subset abundance is also subject to turnover rates.’

For the LPS response finding, we have previously used no background gene set, which we have now corrected and found that the LPS response pathway is no longer enriched among the associated genes if the appropriate background gene set is used. We have thus dropped this initial finding from the manuscript, which in any case has not been crucial for our main discovery.

Minor comments

6. The authors have considered as candidate genes those genes that carry “a variant in this gene expected to severely alter the protein or the transcript” in at least one founder strain. Can the authors verify whether these genes are under weaker negative selection than all other genes?

This may affect the comparison between non-cell-intrinsic genes and cell-intrinsic genes, if their proportions are different in the strongly constrained genes that are not tested.

We have now removed this restrictive criterion (see also the response to Comment #2), and we are considering all immune genes with a variant in an exonic region. While with the criteria presented in the initial version of the manuscript it could theoretically have been that the selection criteria lead to bias against either *cyto-cis* or *cyto-trans* genes, with the current criteria this should not be a concern.

For more clarity on how exactly the selection plays out, we have now included in the Methods section the quantification of how many genes fall in the respective groups – out of 7,965 mouse immune genes identified by the ImmGen consortium, 6,902 genes pass the criterion of having

an exonic variant in at least one founder strain, meaning we only exclude a little more than 13% genes. It is worth pointing out that including these genes into the association analysis would not change anything at all, as without variation there cannot be any association determined. We now exclude them merely for the purposes of reducing multiple hypothesis testing, which in this case we deem justified.

Somewhat related to this point, we have now included the conservation analysis for all immune genes considered and non-immune genes as a baseline in **Fig. 4a**, which are essentially overlying both each other and the conservation data for genes associated in *cyto-cis*. We believe this demonstrates there are no significant biases in our approach that could undermine the conclusion that genes associated in *cyto-trans* are less conserved.

Fig. 4a: Empirical cumulative distribution functions for amino acid conservation scores of genes grouped by whether they were found are associated with immune cell frequencies only exclusively in *cyto-cis* or also in *cyto-trans*. *P*-value for one-sided Kolmogorov-Smirnov test between these two groups is stated. Cumulative distributions of evolutionary conservation for immune genes considered in this study (*Methods*) and for the non-immune genes in the mouse genome are shown for comparison in grey.

7. Important details are lacking for the validation step. Is the direction of effects (observed in the discovery cohort) replicated in the validation cohort? What is the proportion of replicated variants? Can the authors clarify if the validation cohort includes mice that were already present the discovery cohort (Extended Data Table 1)? If so, the analysis should be redone, as a replication cohort should be independent from the discovery cohort.

We thank the reviewer for improving this technical aspect of our work. We indeed have previously not used directionality of effect as a criterion for validation. We agree with the reviewer that this too often is omitted and that this validation criterion certainly should become the gold standard in the field. We have now included the replication of directionality as a criterion to be validated using the validation cohort. While this has resulted in a decrease in the number of associations validated, thanks to these more stringent criteria, the main result in **Fig. 4a**, describing the significant sequence conservation differences between *cyto-trans* genes and other gene sets, came forward yet more clearly.

We also thank the reviewer for the request to clarify the composition of the validation cohort used in the actual analysis. The validation analysis now and also in the initial manuscript excluded mice strains already present in the discovery cohort. We have now amended the presentation as well as the relevant Extended Data Tables.

8. The analysis named “Signal propagation” is unclear. How did the authors use the associated clusters? Did they include clusters when performing enrichment tests? The way associated clusters are defined (i.e., median LOD score of genes in the cluster for the given cell type is at least 40% of the highest median calculated for that cluster) seems arbitrary. Can the authors justify the approach?

Using correlated traits for signal propagation is an established principle, and while many flavours of implementation exist (e.g. as in refs. 50-52), the unifying rationale behind this is that while the determination of association of a single trait for given gene might be marred in noise, if there is a gene cluster with associations correlated across traits, the correlation allows sidelining the noise, which by definition is not correlated across traits.

The 40% threshold was chosen empirically based on the distribution of the relative association scores, which has a trough at 0.4 in between two peaks. This explanation is now included in the *Methods* and the plot of the distribution is now included as **Extended Data Figure 3b**.

Extended Data Figure 3b: Distribution of relative median LOD scores for gene clusters during signal propagation. Genes found significantly associated to at least immune cell type were clustered based on their pattern of association across all the assayed cell types. For each cluster, median LOD score for each cell type was calculated and then divided by the maximum value of this quantity across all cell types. The distribution of the resulting values is plotted, with two clear peaks – all gene cluster-cell type pairs with median LOD score above 0.4 were considered associated.

The clusters were created using *k*-means clustering with the value of *k* determined by the silhouette method, solely for the purpose of Signal propagation. The clusters were not used for

any other analysis, including functional enrichment. We have now included this statement in the *Methods*.

9. Please avoid terms that are uncommon and/or not properly defined. What do the authors mean by “quantitative homeostasis”, “useful variation”, “gain advantage in evolvability”?

We have now replaced ‘*quantitative homeostasis*’ with the more standard term of ‘*homeostatic balance*’, ‘*useful variation*’ with plain ‘*variation*’ and we dropped the convoluted wording ‘*gain advantage in evolvability*’ altogether.

10. Please replace “fewer purifying constraints” by “weaker negative selection”, which is more conventional.

The wording has been replaced as suggested.

11. First paragraph of the introduction: To my knowledge, there is no evidence that “more complex” organisms evolve at the slower evolutionary rate than “less complex” organisms. Please clarify.

We second the opinion of the reviewer that it is not known that complex biological systems would evolve more slowly than the simple ones. However, for complex artificial systems, this is known. Early studies in evolutionary computation (as reviewed in ref. 9) have shown that for complex tasks, quasi random changes did not lead to a successful algorithm. Instead, mutations needed to have a certain structure in order to produce effective solutions. It is presumably this feature, the ability to produce variation of suitable structure, also termed evolvability, that complex biological systems have acquired, effectively resulting in rates of evolution comparable to simple systems. We have now amended the statement in question to make it clear that we were talking here about artificial, not biological systems (Lines 38-45):

‘A key precondition for evolution by natural selection is the availability of suitable variation in natural populations. Early studies in evolutionary computation have shown that increased complexity also increases the probability that random mutations produce pleiotropic effects negatively affecting fitness¹⁰. In other words, complex systems have a larger potential for getting trapped in local fitness optima. Thus, for the Darwinian process of evolution through mutation and selection to work in complex biological systems, the systems need to have evolved evolvability – an architecture such that mutations are likely to result in more adaptive phenotypes².’

12. Please rephrase “This suggests that the observed variation in immune profiles is likely to possess adaptive value in face of bacterial threats.” This is an overstatement; a gene enrichment is not suggestive of any “adaptive value”.

We removed this statement altogether due to improvements in technical aspects of our work (cf. our response to the Comment #5).

13. The statement “while the genetic determinants of inner cell life have been fixed early in the evolution to ensure robust functioning of the individual immune cell types” is highly speculative and questionable. Note that fixation is rare in genes under strong

negative selection. Similarly, the statement “the interaction between different immune cell types provides a flexible phenotypic space where mutations can produce selectable variation without much detriment.” Is highly speculative. The authors provide no evidence that such variation will be adaptive.

This statement was indeed intended to be speculative, but we admit we failed to clearly indicate that this is a speculative attempt at conceptualisation. We have now clearly indicated that we are writing in speculative terms:

‘Conceptually, one can speculate that while the genetic determinants of inner cell life might have been arranged early in the evolution to ensure functioning of the individual immune cell types, the interaction between different immune cell types provides a phenotypic space where mutations can continue to produce variation without much detriment.’

14. Several figure panels are not described or discussed in the text (Figs. 2b, 3c, etc.).

We have now made sure that each respective panel is discussed in the main text.

15. Why does the LODscore significance threshold vary from one cell type to another? Sample size? Please clarify in the Figure 3b legend.

We have now improved the presentation of **Fig. 3b** also thanks to the feedback from other reviewers and the LOD score threshold is no longer depicted in this confusing manner.

Figure 3b: Example of data used to classify associations as *cyto-cis* or *cyto-trans*. Gene expression determined by ImmGen consortium, and log odds (LOD) scores for association with immune cell frequencies are shown for a selected gene, *Arhgef37*. Black dashed line indicates gene expression threshold used for binarization of gene expression (*Methods*), asterisks denote significant associations as determined by a permutation test (*Methods*). Coloured dashed boxes highlight associations in *cyto-cis* and *cyto-trans*.

The LOD scores significance thresholds were determined by bootstrapping, and as such these are dependent on the variation in the frequency of each respective cell type in the mouse cohort. This is explained in the main text, with reference to **Extended Data Fig. 3a**, where this is explained in more detail.

Extended Data Figure 3a: False discovery rate (FDR) as a function of threshold for association as inferred from permutation analysis (*Methods*) for each cell type. Red dashed line indicates the FDR of 5% used to determine the cell type-specific LOD score threshold used to find associated genes in the exploratory cohort.

And, in *Methods* (Lines 389-393): ‘Then, for each cell type, we identified the significant association threshold as one that corresponds to 5% false discovery rate (FDR, **Extended Data Fig. 3a**). We determined the FDR by permutation analysis – we repeatedly shuffled the labels of recombinant mouse strains (replicates were always assigned identical labels) and calculated the strength of the cell type association.’

16. Fig. 4b does not show age estimates but amino-acid conservation score, so it cannot support the statement that “genes associated with the frequency of multiple cell types are evolutionary younger”.

We have now improved the wording in line with the suggestion: ‘We indeed found that genes associated with the frequency of multiple cell types are evolutionarily less conserved than those affecting just one immune cell type.’ (Lines 236-238)

17. Can the authors briefly discuss how variants on the MegaMUGA SNPchip were ascertained and if this ascertainment can bias their results?

The MegaMUGA chip has been ascertained specifically to fit the Collaborative Cross cohort (Morgan et al., G3, 2016, PMID: 26684931), so the ascertainment bias should have been minimised within experimental limits, although it certainly cannot be avoided. Ascertainment bias could in principle influence whether certain groups of genes (e.g. evolutionary more recent genes) are found to be associated at all, and in this way influence the result of functional enrichment analysis and the distribution of number of associated genes found across different cell types. However, our main result is comparative, finding that genes associated in *cyto-trans* are less conserved than those associated in *cyto-cis* – we see no plausible mechanism of how ascertainment bias could lead to preferentially finding association in weakly conserved genes for *cyto-trans* genes but not for *cyto-cis* genes; the expression or non-expression of a certain gene in a particular cell type is unrelated to ascertainment bias.

18. According to EU legislation (2010/63/EU), the term “euthanasia” should be avoided, as it implies that mice were suffering at the time of the killing, and should be replaced by “killing” (<https://www.nc3rs.org.uk/3rs-resources/euthanasia>).

The wording was changed as suggested.

19. Can the authors illustrate with a Suppl. Figure how the cell count data changes after within-sample, over-time normalization?

We thank the reviewer for requesting that we describe this technical aspect in more detail. The initial measurements for the current study were performed at a time when CyTOF was a new technology, and we faced technical challenges, most notably the instability of CyTOF signal after a longer acquisition time. As a default, the samples were measured for 10 minutes. In some cases, after the initial period of stable signal, some of the channels exhibited drift (see an example plot below, y-axis CyTOF signal, x-axis time in ms). To address this technical issue during post-processing, we truncated the acquisition records at the moment when any of the channels drifted by more than 5% from its mean since the start of the acquisition (see red dashed line in the example plot for the truncation point). Strictly speaking, this is a very straightforward data filtering rather than normalisation, hence the original formulation in the *Methods* was misleading. We have now corrected and clarified the corresponding section in the *Methods*.

R2R Figure 2: Stability of CyTOF signal over time. In case the CyTOF signal in any of the channels drifted more than 5% over time, the acquired data was truncated during post-processing as shown in the example data above. Red dashed line indicates the time point at which the data was truncated; only data acquired for this sample before this time point were considered for further analysis.

Referee #3:

The article aims to demonstrate that genetic factors influencing the frequency of immune cell types are expressed within the corresponding cell types, thus exhibiting cell-intrinsic functions (cyto-cis genes). Additionally, the study reveals that certain genes impacting the frequency of a particular cell type are not expressed within these cells but rather in other cells within the immune system, indicating a function in trans (cyto-trans genes).

We thank the reviewer for the overall useful and constructive suggestions on our work. We would like to add to this summary that our main result goes beyond identifying *cyto-cis* and *cyto-trans* genes, by showing that the evolution of *cyto-trans* genes is less constrained than that of the *cyto-cis* genes and, consequently, that interactions between cell types are important for the immune system's evolvability. We suggest that this observation, coupled with the implied need of the immune system to evolve rapidly, could in part explain the modular organisation of the immune system into many different cell types.

I have several questions/ concerns listed in the following paragraph

Abstract

1. Lane 4 The authors wrote:

"The immune system is one of the fastest-evolving components of mammalian genomes. They cite Shultz AJ and Sackton TB immune genes serve as hotspots for shared positive selection across birds and mammals Elife 2019.

I am curious about the gene overlap between The two studies.

Shultz & Sackton used 39 avian genomes and identified 11,248 bird orthologues; they found 1,375 of them positively selected. In our study, we considered 6,902 mouse immune genes and found & validated 271 of them associated with the frequency of at least one immune cell type in the mouse bone marrow. Out of the 1,375 bird genes and 271 mouse genes, not all of them have annotated orthologues in the other species. Out of the 576 bird genes and 161 mouse genes that do, the gene overlap is 24 genes. This overlap is not statistically significant ($p_{\chi^2} = 0.36$).

We think the reason for this might be that Shultz & Sackton are looking at a different evolutionary timescale and a different biological question than we are. They used 39 bird species for their analysis; having significant overlap between genes found to be positively selected during this evolutionary timespan and in our association study would mean that, i) the evolution of regulators of the frequency of the immune cells in the mouse bone marrow happened preferentially during the evolutionary timespan of bird evolution and that, ii) the evolutionary process in the birds and mammals with regard to bone marrow was driven by similar factors. This is unlikely given anatomical differences between mammals and birds, whose marrow is substantially different.

2. In Lane 16, (and also in the discussion session) the authors state, 'Our findings suggest that interactions between different components of the immune system provide a phenotypic space where mutations can produce selectable variation without significant detriment, offering a potential resolution to the robustness-evolvability conundrum within the context of the immune system.' They reference the work of Wagner and colleagues.

However, in a more recent paper, the same research group contradicts this statement. They argue that robustness actually increases evolvability (Zheng J, Guo N, Wagner A. 'Selection enhances protein evolvability by increasing mutational robustness and foldability.' Science. 2020 PMID: 33273072)

We regret that our wording gave the impression that we contradict these recent results of Wagner and colleagues, while in fact, we essentially build on them. What we meant to say right from the start was that in their recent work, indeed Wagner and colleagues showed that increased mutational robustness essentially equates to increased evolvability. In our work, we show that genes associated with immune traits in *cyto-trans* are under weaker negative selection, i.e. mutations in these genes would not be expected to cause strong negative phenotypic effects, endowing the immune system with increased robustness towards mutations and hence, evolvability. We have now improved the wording both in the *Abstract* and in the *Discussion* so as to not give this misleading impression (Lines 291-301):

'While variation with strong positive or negative fitness effects have appreciated roles in the evolution of biological species, the importance of near-neutral variation has only lately emerged as an important consideration for the capacity of biological systems to evolve. Recent work on transcription factor and protein structure evolution^{2,7,58} has found that increased capacity for near-neutral variation, i.e. increased mutational robustness, paradoxically facilitates evolution by supporting genetic diversity. This has solved the apparent conflict between evolvability and robustness⁸ – the seemingly contradictory requirement that phenotypes be significantly altered by genetic changes to allow selection of fitter phenotypes, but not be altered too much to endure frequent mutations without harm. Our results offer an appealing extension of this evolvability principle in the context of the immune system – an enhancement of the near-neutral genetic diversity is achieved through modularity.'

Main text

3. Lane 2 the authors wrote : While in simple systems, random mutations are likely to produce beneficial phenotypes at a non-negligible frequency.

The authors need to provide reference here.

We thank the reviewer for pointing out the cumbersome formulation we used. We have now removed this part of the sentence and the whole statement was amended to reflect more precisely that we were talking about artificial rather than biological systems here (Lines 38-45):

'A key precondition for evolution by natural selection is the availability of suitable variation in natural populations. Early studies in evolutionary computation have shown that increased complexity also increases the probability that random mutations produce pleiotropic effects negatively affecting fitness¹⁰. In other words, complex systems have a larger potential for getting trapped in local fitness optima. Thus, for the Darwinian process of evolution through mutation and selection to work in complex biological systems, the systems need to have evolved evolvability – an architecture such that mutations are likely to result in more adaptive phenotypes².

4. Lane 8: the authors wrote *The immune system is a complex system intricately involved in the battle against pathogens, making it a prime target for the process of natural selection.*

It's worth noting that the immune system also plays a role in maintaining homeostasis.

We agree that this was an important omission and we have rectified it: *'The immune system is a complex system intimately engaged in maintaining homeostasis and the struggle against pathogens, which makes it a prime target for the process of natural selection.'* (Lines 46-48).

Could the author clarify if the cell type corresponds to the module? This is what I understand from the text.

We apologise for the lack of clarity in the initial manuscript as to whether the evolutionary modules that interact with each other to increase evolvability necessarily correspond to cell types. We also thank the reviewer for the stimulus to revisit our results again in light of this question. Our main result suggests that cell types indeed represent such evolutionary modules, but we now believe that even groups of cell types could form yet another layer of evolutionary modules. In fact, our result in **Fig. 4b** shows that genes associated with multiple cell types, i.e. potentially eliciting changes coordinated across cell subsets, are also under weaker negative selection than genes affecting just a single cell subtype ($p_{KS} = 1.1 \cdot 10^{-5}$). The fact that genes coordinating across multiple cell types are also a potent source of evolvability, suggest the existence of 'supermodules' encompassing multiple cell types.

Figure 4b: Empirical cumulative distribution functions for amino acid conservation scores of genes grouped by whether or not they are associated to more than one immune cell type. *P*-value for one-sided Kolmogorov-Smirnov test between these two groups is stated. **c.** Empirical cumulative distribution functions for amino acid conservation scores of genes associated with frequencies of multiple immune cell types grouped by whether they were found associated with immune cell frequencies only in cyto-cis or also in cyto-trans.

We have now written this explicitly in the *Discussion* section (lines 306-313):

'A cell, the atomic unit of life, is well suited to correspond to a module in the evolutionary process of the immune system, consistent with cyto-trans genes acting as an important source of evolutionary novelty. Nevertheless, our observation that genes associated with frequencies of multiple cell types, and thus possibly coordinating between them, are also an important source of evolvability irrespective of the cyto-trans phenomenon, suggests the existence of multiple layers of modularity. Such staged modular design, where modules are iteratively combined into yet more complex modules, has been previously proposed to be one of the unifying features of evolved complex systems, including those of human origin⁹.'

5) Lane 20

"The authors cite many articles looking at the genetic determinants of immune profiles (DOI: 10.1016/j.cell.2020.08.008). I think it is important to investigate the overlap between the candidate genes found in their study and the studies conducted by other investigators.

We thank the reviewer for this important point and have followed up on it. We have now tested our result that *cyto-trans* genes facilitate the evolution of immune traits in the context of a different organism (human) and a different organ (peripheral blood), using also data from the article suggested by the reviewer.

We performed an analogous analysis of the evolutionary conservation of *cyto-trans* vs *cyto-cis* associated genes as we did for immune cell frequencies in the mouse bone marrow, but this time for immune cell frequencies in the human peripheral blood. We obtained published data from two human studies of immune cell frequencies in the blood (Vuckovic et al., 2020, *Cell*, PMID: 32888494, Chen et al, 2020, *Cell*, PMID: 32888493) In total, they found 1046 genes to be associated with abundance of at least one of these cell types. We juxtaposed these genes with the gene expression data from Human Protein Atlas. Here as well we found that genes associated with cell types in which they were not expressed (*cyto-trans*, 358 genes) were significantly less conserved than those expressed in the associated cell type ($p_{KS} = 6.0 \cdot 10^{-10}$).

Our results replicate what we observe in mouse, namely that genes associated in *cyto-trans* exist, that they face weaker negative selection than *cyto-cis* genes and hence, they are a source of evolvability. This result greatly strengthens our analysis, so it is now part of the manuscript as **Fig. 4d**:

Figure 4d: Empirical cumulative distribution functions for amino acid conservation scores of genes found associated with immune cell frequencies in the human blood by previous studies. Genes are grouped by whether they were found associated with immune cell frequencies only in *cyto-cis* or also in *cyto-trans*.

With respect to investigating the exact gene overlap, we found limited utility in this analysis, presumably because our study was not only performed in a different organism, but also in a different organ. Below is a Venn diagram illustrating the overlap of associated genes, while considering only those mouse genes that have a human orthologue and *vice versa*.

R2R Figure 3: Venn diagram of mouse genes associated with the immune profiles in the bone marrow found in this study and human genes associated with immune profiles in the blood found in previous studies. Only mouse genes with a human orthologue and human genes with a mouse orthologue were considered. Human data were pooled from Vuckovic et al., 2020, *Cell* and Chen et al., 2020, *Cell*.

6) Page 4 variation in immune profile is polygenic

""The author wrote, 'We consider only genes that are expressed in at least one immune cell type.' However, in their study, they demonstrate that genes expressed in trans affect the frequency of different cell types. I believe that by restricting their consideration to genes expressed in at least one immune cell type, they may overlook some potential gene candidates (that are not expressed in immune cell type) Maybe I miss something : please clarify

We agree with the reviewer and thank him for noting this oversight. While every study has certain limitations, in the current version of the manuscript we consider all genes with tentative immune function. We believe this addresses the concern.

We have now updated *Methods*, section *Filtering of Loci* as follows: ‘We included only loci that passed both of the following criteria: First, we chose loci located within a broad set of genes that were identified previously by the ImmGen consortium (7,965 genes) as having a potential function in the immune system^{46,47}. Second, we chose loci for which at least one of the CC founder strains harbours an exonic sequence variant.’

7) It would also helpful to have the ratio of coding versus non coding variation that impact the phenotype and compare their data with DOI: 10.1016/j.cell.2020.08.008

In our study, we have made the decision to favour interpretability over comprehensiveness. Thus, we have made the choice not to explore the variation in the intergenic and intronic variants which are generally difficult to link to particular genes. To illustrate this point, Borsari et al. (Genome Research 2021, PMID: 34290042) found that intronic enhancers active in the blood actually control the gene containing the intron less frequently than some other gene. Although our analysis presently includes some non-coding variation, e.g. those variants in the 5’UTR (see *Methods*, section *Filtering of Loci*), the choice we made for the purpose of interpretability of our work precludes the comparison suggested by the reviewer.

8) Generally, do cyto-trans genes code for cytokines or other soluble elements? In other words, are some of these genes already known to function in trans?" More specifically, is Bex1 known to act in trans?

Thanks to Reviewer #2, we have now implemented a more stringent approach to gene validation (incorporating more stringent thresholds and a requirement that the effect size direction be observed in the validation cohort. This resulted in fewer genes associated at this more stringent threshold, but also in a strong increase in observed difference in evolutionary conservation between genes associated in *cyto-trans* vs. *cyto-cis* – key to identifying the role of *cyto-trans* genes in the evolutionary process. As a consequence though, *Bex1* is no longer among the associated genes and we have chosen a different example gene to be depicted in Fig. 3b.

Figure 3b: Example of data used to classify associations as *cyto-cis* or *cyto-trans*. Gene expression determined by ImmGen consortium, and log odds (LOD) scores for association with immune cell frequencies are shown for a selected gene, *Arhgef37*. Black dashed line indicates gene expression threshold used for binarization of gene expression (*Methods*), asterisks denote significant associations as determined by a permutation test (*Methods*). Coloured dashed boxes highlight associations in *cyto-cis* and *cyto-trans*.

Separate but related to this, we wish to clarify that in our manuscript we talk about interaction between cell types rather than cells. This semantic distinction is important – as our use of the word interaction is not in the narrow sense of ‘physical interaction between cells’, but rather in its broader non-restricted meaning which includes e.g. genetic interaction, chemical interaction or indirect interactions. Correspondingly, we discuss the interactions between cell types in the sense of how cells of one cell type influence the count of another cell type and ‘interaction’ here is meant in this sense of the word, as we stated also in our original manuscript (now lines 186-189). ‘..., the relevant immune cell types must be influenced by these variants through some form of interactions with cell types that express the protein, possibly mediated by ligands, metabolites, yet other cell types, or in a connection between a cell and its precursor.’

We have now also performed pathway enrichment analysis for the *trans*-associated genes specifically using all associated genes as a background set (**Extended Data Fig. 4b**). A number of signalling pathways came enriched here, congruent with the role of these genes in regulating and/or mediating interactions between cell types.

Extended Data Figure 4b: Bar-plot of functionally enriched categories in *cyto-trans* genes relative to all associated genes. Functional groups that were found enriched using Ingenuity Pathway Analysis terms with $p < 0.05$ from Fisher’s exact test are shown.

9) Page 9: *Cyto-trans* genes and evolution

I believe this subtitle is somewhat misleading. The authors demonstrate that cyto-trans gene evolution is weakly constrained, but they do not necessarily show that these genes actively facilitate evolution.

We agree that we indeed show that *cyto-trans* gene evolution is weakly constrained. However, we believe that the logical consequence of this observation is that these genes associated with important phenotypic traits in *cyto-trans*, exactly because their evolution is weakly constrained, increase the evolvability of the respective traits. In other words, they facilitate (make easier to happen) the evolution of these traits. The central idea here is that phenotypically near-neutral variation, also known as mutational robustness, increases evolvability (Payne & Wagner, *Science* 2014, PMID: 24558158). Demonstrating that the evolution of *cyto-trans* genes is weakly constrained also means that these genes can offer more near-neutral variation than genes associated in *cyto-cis*, and thus they increase the evolvability of associated traits. Since increasing evolvability represents one of the ways of facilitating evolution, we believe it was not misleading to use this term. We agree though with the reviewer that stripped to essentials, we show that *cyto-trans* gene evolution is weakly constrained. We have now amended the subtitle to this end: '*Cyto-trans genes face weaker negative selection*', and we have now tried to make this logical link between increasing near-neutral variation and facilitating evolution clearer in the Discussion (lines 282-287 and 291-296).

'Notably, we found that the coding sequences of cyto-trans genes have been under weaker negative selection during vertebrate evolution than those of the genes found acting in cyto-cis only. This implies that the genetic determinants of interactions between different immune cell types are, when mutated, more amenable to produce near-neutral variation in immune cell frequencies compared to genes involved in internal cell regulation.' ... *'While variation with strong positive or negative fitness effects have appreciated roles in the evolution of biological species, the importance of near-neutral variation has only lately emerged as an important consideration for the capacity of biological systems to evolve. Recent work on transcription factor and protein structure evolution^{2,7,58} has found that increased capacity for near-neutral variation, i.e. increased mutational robustness, paradoxically facilitates evolution by supporting genetic diversity.'*

Reviewer Reports on the First Revision:

Referees' comments:

Referee #1 (Remarks to the Author):

The authors have sufficiently addressed my (minor) concerns. I really enjoyed the manuscript and support its publication.

Referee #2 (Remarks to the Author):

Dubovik and colleagues have satisfactorily addressed my comments. Statements regarding evolvability are better justified and the vocabulary is overall straightforward. The replication in humans greatly strengthens the authors' conclusions. I still believe that some results are slightly over-interpreted, but the study raises interesting hypotheses.

I have remaining, minor comments.

- End of the introduction: The authors state that "those genes not expressed in the cell type whose variation they influence, have been preferred as the source of variation by the recent evolution in vertebrates". I suggest instead: "..., have accumulated functional variation under weak negative selection, supporting a role of inter-cellular interactions in immune system evolvability."

- In the text, the authors refer to 30 recombinant inbred strains and cite Fig. 1a, where 30+24 RI strains are shown. Please clarify Fig 1a legend accordingly.

- Fig. 1c shows a triallelic variant, which is less common. Why not showing a biallelic variant?

- When stating "Functional enrichment of this larger set confirmed the role of cell intrinsic functions", I presume authors refer to Fig. 2c rather than 3c.

- It is unclear why certain cell types have not been analysed. The Methods report 11 cell types, the main text lists 9 cell types (hematopoietic stem cells, NK cells, CD8+ T cells, CD4+ T cells, total B cells, pro-B cells, late B cells, granulocytes and monocytes), and most results are reported for 8 cell types (e.g., NKs are absent from Fig. 2b). Please clarify in the Methods and Figure legends. Also, please clarify which cell types were measured in the replication sample.

- Fig. 3d may be further commented. Why are most associated genes observed in late B cells and monocytes? Why are cyto-trans genes expressed mainly in HSC?

Author Rebuttals to First Revision:

General remarks to all reviewers:

We thank all the reviewers for their constructive feedback that helped strengthen our work. We address the remaining minor issues below.

Referee #2:

Dubovik and colleagues have satisfactorily addressed my comments. Statements regarding evolvability are better justified and the vocabulary is overall straightforward. The replication in humans greatly strengthens the authors' conclusions. I still believe that some results are slightly over-interpreted, but the study raises interesting hypotheses.

I have remaining, minor comments.

- End of the introduction: The authors state that "those genes not expressed in the cell type whose variation they influence, have been preferred as the source of variation by the recent evolution in vertebrates". I suggest instead: "..., have accumulated functional variation under weak negative selection, supporting a role of inter-cellular interactions in immune system evolvability."

We have replaced the respective part with wording closely matching the reviewer's suggestion.

- In the text, the authors refer to 30 recombinant inbred strains and cite Fig. 1a, where 30+24 RI strains are shown. Please clarify Fig 1a legend accordingly.

We have corrected the caption in Fig. 1a to reflect that 30 strains were assayed in the exploratory cohort.

- Fig. 1c shows a triallelic variant, which is less common. Why not showing a biallelic variant?

We agree with the reviewer that it makes sense to simplify this figure and are now conceptually showing a biallelic variant.

- When stating "Functional enrichment of this larger set confirmed the role of cell intrinsic functions", I presume authors refer to Fig. 2c rather than 3c.

We thank the reviewer for catching this mistake which is now corrected.

- It is unclear why certain cell types have not been analysed. The Methods report 11 cell types, the main text lists 9 cell types (hematopoietic stem cells, NK cells, CD8+ T cells, CD4+ T cells, total B cells, pro-B cells, late B cells, granulocytes and monocytes), and most results are reported for 8 cell types (e.g., NKs are absent from Fig. 2b). Please clarify in the Methods and Figure legends. Also, please clarify which cell types were measured in the replication sample.

We regret this unclarity. We have determined experimentally 9 cell types as stated in the main text. In an earlier version of the work, we have estimated the frequency of two cellular subsets through a statistical learning approach involving some of the 9 subsets, but we have later abandoned this approach due to the complexity of interpretation. For NK cells, we could not determine any genetic association in our study design, which is now explicitly stated in the legend to **Fig. 2b**. In the validation cohort, the same cell types were measured as in the exploratory cohort and this is now explicitly stated in the *Methods* section 'Validation of gene-trait association'.

- Fig. 3d may be further commented. Why are most associated genes observed in late B cells and monocytes? Why are cyto-trans genes expressed mainly in HSC?

These are indeed intriguing phenomena deserving further study. While the former question targets something potentially explainable by technical aspects of our study, e.g. the extent of variation in these subsets in our particular cohort, the latter question is most likely about a genuine biological phenomenon. One reason why HSCs might express many *cyto-trans* genes might be that in the HSCs, they control the rate with

which the daughters of HSCs differentiate into other cell types. We have now reflected this possible explanation in the wording of the results section:

“While many cyto-trans genes were expressed in HSCs, suggesting the control of abundance of a specific cell type from upstream in the differentiation lineage, overall, cyto-trans genes create a complex web of interactions between cell types (Fig. 3d).”